# Pathway metabolite ratios reveal distinctive glutamine metabolism in a subset of proliferating cells

Nancy T Santiappillai [1,4], Yue Cao[2,3], Mariam F Hakeem-Sanni[1], Jean Yang [2,3], Lake-Ee Quek[2] & Andrew J Hoy [1✉]

## Abstract

**Large-scale metabolomic analyses of pan-cancer cell line panels have provided significant insights into the relationships between metabolism and cancer cell biology. Here, we took a pathway-centric approach by transforming targeted metabolomic data into ratios to study associations between reactant and product metabolites in a panel of cancer and non-cancer cell lines. We identified five clusters of cells from various tissue origins. Of these, cells in Cluster 4 had high ratios of TCA cycle metabolites relative to pyruvate, produced more lactate yet consumed less glucose and glutamine, and greater OXPHOS activity compared to Cluster 3 cells with low TCA cycle metabolite ratios. This was due to more glutamine cataplerotic efflux and not glycolysis in cells of Cluster 4. In silico analyses of loss-of-function and drug sensitivity screens showed that Cluster 4 cells were more susceptible to gene deletion and drug targeting of glutamine metabolism and OXPHOS than cells in Cluster 3. Our results highlight the potential of pathway-centric approaches to reveal new aspects of cellular metabolism from metabolomic data.**

**Keywords** Metabolomics; Metabolic Pathways; Cancer Cell Lines; Glutamine Metabolism; Glucose Metabolism
**Subject Categories** Cancer; Metabolism

## Introduction

Cancer cells are characterized by dynamic plasticity of nutrient utilization that supports tumor growth and survival (Altea-Manzano et al, 2020; Fendt et al, 2020; Pavlova et al, 2022). Our understanding of the role of metabolism in cancer cell biology has predominantly arisen from studies taking cancer type-specific and/ or metabolic pathway-focused approaches (for example, (Hensley et al, 2016; Kamphorst et al, 2015; Wang et al, 2023)).

More recently, several publications have reported the outcomes of high-throughput metabolomics using pan-cancer cell line panels, such as the NCI-60 (60 cell lines, (Jain et al, 2012; Ortmayr et al, 2019) and CCLE panels (928 cell lines (Li et al, 2019), CAMP (988 tissue samples (Benedetti et al, 2023), and other panels containing 180 cell lines (Cherkaoui et al, 2022), and 173 cell lines (Shorthouse et al, 2022). These studies primarily aimed to identify links between cancer cell metabolic phenotypes and transcriptional regulation (Benedetti et al, 2023; Ortmayr et al, 2019), or genetic alterations and dependencies (Li et al, 2019; Mullen and Singh, 2023) that were associated with drug-sensitivities (Shorthouse et al, 2022). Notably, Cherkaoui and colleagues (2022) took a top-down approach by clustering the metabolome acquired by untargeted metabolomics across 49 KEGG metabolic pathways of 180 cancer cells. Pathway activity was determined using principal component analysis to quantify ubiquitous and coordinated changes in multiple metabolites within each pathway, with the PC1 scores used as proxy of pathway activity for each cell line. From this approach they identified only two clusters that were defined by either high carbohydrate metabolic activity or high aerobic mitochondrial activity, that was associated with epithelial or mesenchymal status, respectively (Cherkaoui et al, 2022).

Here, we took a different approach to determine whether cancer cells can be clustered into subtypes of metabolic pathway activity based upon only high-flux metabolic pathways in a smaller pan-cancer panel of proliferating cells. Based on metabolite concentrations, we initially identified four clusters of cells, but this approach failed to provide insights into pathway differences between these clusters. To overcome this issue, we transformed our metabolite concentration data into pathway-centric ratios. We believe that this conceptual innovation, organizing our data into pathways rather than as individual metabolites, would more directly identify differences in pathway behavior. Our approach differed to Cherkaoui and colleagues (2022) as we retained all quantified metabolites, including those reactions and corresponding substrate or product present in multiple metabolic pathways, such as ATP, NADH, glucose-6-phosphate. Our approach resulted in the formation of five clusters of cells that displayed different ratios of

[1]School of Medical Sciences, Charles Perkins Centre, Faculty of Medicine and Health, The University of Sydney, Sydney, New South Wales 2006, Australia. [2]School of Mathematics and Statistics, Charles Perkins Centre, Faculty of Science, The University of Sydney, Sydney, New South Wales 2006, Australia. [3]Sydney Precision Data Science Centre, The University of Sydney, Sydney, New South Wales 2006, Australia. [4]Present address: Radiology, and Molecular Pharmacology Program, Memorial Sloan Kettering Cancer Center, New York, NY, USA. ✉E-mail: andrew.hoy@sydney.edu.au

metabolites of glycolysis, pentose phosphate pathway, pyruvate-TCA cycle, proline metabolism, serine metabolism, glutamine metabolism, and methionine metabolism. Of these five clusters, we used a combination of techniques to show that cells in Cluster 4 had higher ratios of TCA cycle metabolites when normalized to pyruvate and produced more lactate, despite lower glucose and glutamine consumption, and greater OXPHOS activity than Cluster 3 with low TCA cycle metabolite ratios. These differences were, in part, explained by increased glutamine cataplerotic efflux and glutaminolysis. These phenotypes were supported by in silico analyses of pan-cancer loss-of-function and drug sensitivity screens to show that cells in Cluster 4 were more susceptible to gene deletion and drug targeting glutamine metabolism and OXPHOS compared to the cells in Cluster 3. These results highlight the benefit of converting metabolite levels into pathway-based ratios as a starting point for gaining insights into cellular metabolic activity.

# Results

## Targeted metabolomic profiling of high flux pathways in cell lines from 11 tissue origins

We used LC-MS/MS to quantify the concentrations of 50 metabolites that are members of high flux pathways, including central carbon (glycolysis, TCA cycle, pentose phosphate pathway) and amino acid metabolic pathways, in 57 adherent cell lines (49 tumor- and 8 normal epithelial-derived) from 11 cancer types cultured in basal media conditions (Table EV1). Samples were generated in triplicate across 6 batches, including control cell lines for batch correction, ensuring consistency throughout the complete dataset (Fig. 1A). The concentrations of the 50 metabolites quantified are reported in Dataset EV1.

To group cell lines according to shared metabolite profiles, K-means clustering based on Pearson's correlation coefficient was carried out on the batch-corrected metabolite concentration dataset (Fig. 1B). Given the known limitations of K-means clustering (Ren et al, 2015), we used a combination of gap statistics (Tibshirani et al, 2002), the elbow (Thorndike, 1953) and silhouette methods (Rousseeuw, 1987) to determine the optimum number of clusters, which was four. We observed that the culturing conditions, tissue type (normal epithelial vs. tumor), cancer type, tissue origin, or the mutation status of common oncogenic metabolic drivers did not explain how cells lines are grouped together (Cairns et al, 2011; Jia et al, 2008; Jones and Thompson, 2009; Oermann et al, 2012; Vousden and Ryan, 2009) (Fig. 1C). All clusters contained tumor and normal cell lines from different tissue origins (Fig. 1B), suggesting that signatures based on metabolite levels were not sufficiently distinctive between cancer and normal cell lines. There were some instances where cell lines from the same tissue origin clustered together, such as Cluster 3 that was enriched with prostate cancer cells and Cluster 4 with endometrial cancer cells (Fig. 1B), which has been observed in another pan-cancer metabolome study (Shorthouse et al, 2022). Despite the identification of heterogeneous clusters of cells based upon the levels of metabolites of high flux pathways, there was no clear organization of these metabolites into pathways (Fig. 1B). For example, pentose phosphate metabolites (i.e., E4P, 6PG, S7P, hexose phosphate) were distributed through the vertical clusters. This lack of metabolite

clustering into pathways is likely because strong metabolite interactions are often localized at the reaction level (Benedetti et al, 2023). Overall, concentrations of metabolites from high flux pathways failed to cluster cells that could underpin a testable hypothesis centered on differences or shared metabolic pathway activity.

## Analyses of pathway-centric metabolite ratios uncover distinctive metabolic signatures

Next, we took a physiological-based approach and evaluated the hypothesis that cancer cell lines of different tumor origins can be clustered into metabolic subtypes, evident as common pathway activity that is distinct from other clusters. To achieve this, we transformed our metabolomic data of high-flux metabolic pathways by calculating the ratios between an upstream precursor or reactant metabolite (applied as the denominator) and downstream pathway product metabolites (numerator) for each central carbon and major amino acid metabolism pathway, similar to (Benedetti et al, 2023). This pathway-centric transformation was based on the idea that metabolite conversion forms a cascade, and therefore, intrinsic correlations likely exist between reactant and product that provide biologically meaningful insights into pathway activity. For example, glucose is the precursor of glycolysis, and as such, ratios of the abundance of glycolytic metabolites relative to glucose were calculated (Fig. 2A).

We carried out K-means clustering again on the entire metabolite ratio dataset spanning seven metabolic pathways of interest (glycolysis, pentose phosphate pathway, pyruvate-TCA cycle, proline metabolism, serine metabolism, glutamine metabolism, and methionine metabolism), with the number of clusters chosen to be 5 (Fig. EV1A), determined using a combination of gap statistic (Tibshirani et al, 2002), elbow method (Thorndike, 1953) and silhouette method (Rousseeuw, 1987) (Fig. EV1B). These clusters differed from what was identified using metabolite abundances alone (Fig. 1) and were composed of cells from different tissue origins, tissue types, cancer types, mutation status of common oncogenic drivers, and culturing conditions (Fig. EV1C). Like our original 4 clusters, normal cells (e.g., MCF10A mammary epithelial, PNT1 and PNT2 prostate epithelial, PH5CH8 & AML12 hepatocyte, HUE-T & MAD11 endometrial epithelial, and HPDE pancreatic epithelial) were interspersed throughout the 5 clusters, inferring that proliferating cells of non-tumor origins are not metabolically distinct from those of tumor origin.

To assist in interpreting the patterns in the data, the primary heatmap (Fig. EV1A) was separated into individual panels with the cell clusters conserved (Fig. 2). The subset of cells in Cluster 3 had greater glycolysis (Fig. 2A) and pentose phosphate pathway metabolite ratios (PPP; Fig. 2B) but lower TCA cycle (relative to pyruvate; Fig. 2C) and proline metabolism ratios (Fig. 2D) compared to Cluster 2. We identified differences in serine metabolism between cells in Clusters 4 and 5 (Fig. 2E) and less striking differences in glutamine (Fig. 2F) and methionine (Fig. 2G) metabolism between clusters. These patterns suggest that clusters of cells exhibit sufficiently unique metabolic pathway "activity" to be distinctive from other clusters of cells. Together, our approach of transforming metabolite levels into pathway-specific ratios identified groups of cancer cells from different tissue lineages defined by differences in high flux metabolic pathways, not evident by metabolite levels alone.

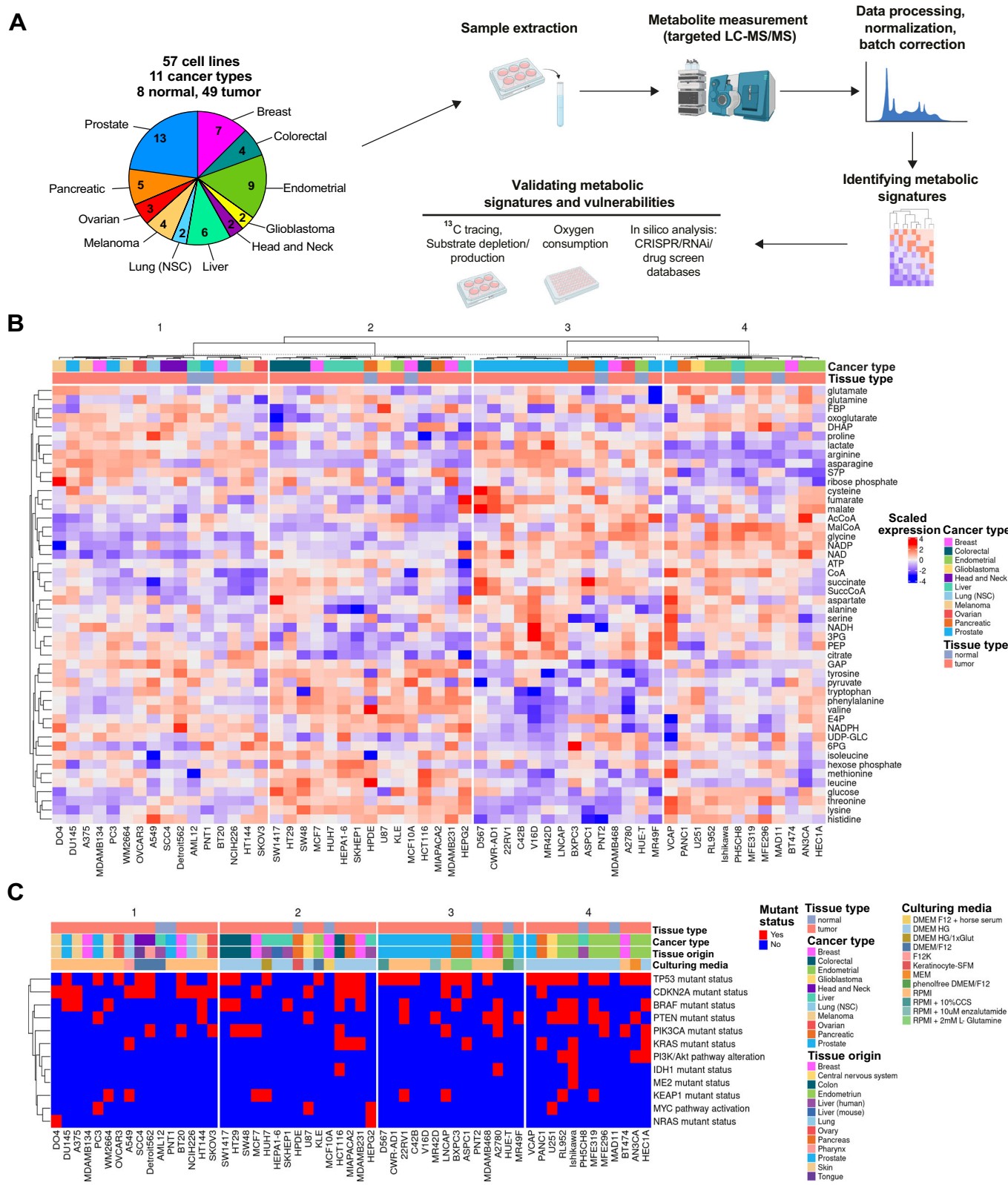

**Figure 1. The metabolome landscape of cells from 11 tissue origin sites.**

(A) Schematic of the workflow for targeted metabolomics profiling of 57 cell lines to identify and validate metabolic signatures. (B) Heatmap of scaled metabolite expression across central carbon and amino acid metabolism within the cell line panel ($n = 57$). Cell lines color coded by cancer type and tissue type. (C) Clusters of cell lines from (B) appended with color coded legends for mutant status of common oncogenic drivers, culturing media conditions, tissue origins, cancer type, and tissue types. Source data are available online for this figure.

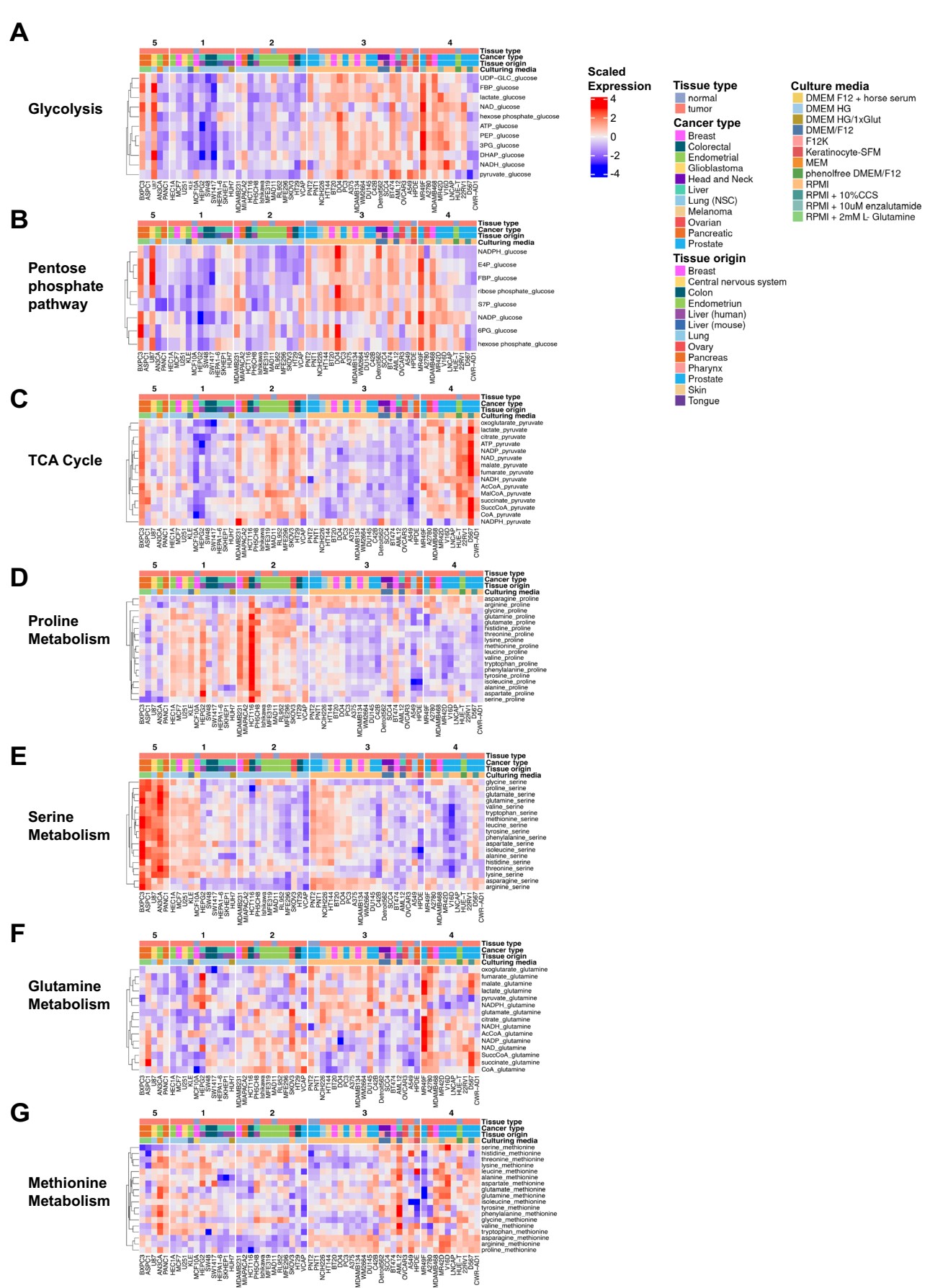

**Figure 2.  Pathway-centric metabolite ratios identify clusters of cells.**

(A–G) Metabolite ratio heatmaps formed by normalization of (A) glycolysis pathway and (B) pentose phosphate pathway metabolites to intracellular glucose, (C) TCA cycle pathway metabolites to pyruvate, (D) proline metabolism metabolites to proline, (E) serine metabolism metabolites to serine, (F) glutamine metabolism metabolites to glutamine, and (G) methionine metabolism to methionine. Heatmaps are slices from Fig EV1. Source data are available online for this figure.

## Differences in TCA cycle metabolite to pyruvate ratios are due to glutamine and not glucose metabolism

To date, irrespective of how the metabolomics data is analyzed, we often cannot infer function/flux, since the increase of an individual metabolite can be due to both an increase in pathway flux and a block in downstream reactions. Here, we sought to determine whether the clusters of cells formed using pathway-centric ratios corresponded to functional differences in metabolic activity. Since the TCA cycle is an essential hub where various pathways converge and is critical for energy and biomass production and cell viability (Spinelli and Haigis, 2018) (Fig. 3A), we chose to contrast Cluster 3 and Cluster 4 from the identified five clusters formed in Fig. 2, as they had distinctive TCA cycle metabolite levels relative to pyruvate (Fig. 3B). Firstly, Cluster 4 cells had greater TCA cycle metabolite levels relative to pyruvate (Multiple unpaired student's t-test corrected for FDR, adjusted $p < 0.05$), except for NADPH (Multiple unpaired student's t-test corrected for FDR, adjusted $p = 0.055$; Fig. 3C). Differences in succinate and malate levels relative to pyruvate were also evident at the cell line level (Fig. 3D).

As pyruvate was used as the denominator to calculate TCA cycle metabolite ratios, we extended our analysis to lactate since pyruvate is converted to lactate by lactate dehydrogenase. Cells in Cluster 4 had a greater lactate-to-pyruvate ratio, a measure of the equilibrium constant of lactate dehydrogenase, than cells in Cluster 3 (unpaired student's t-test, $p = 0.001$; Fig. 3E). There was no difference in the NADH/NAD+ ratio (Fig. EV2A), which are co-factors of lactate dehydrogenase and can influence its activity (Luengo et al, 2021). Thus, the greater lactate to pyruvate ratio in Cluster 4 cells, compared to Cluster 3 cells, was unlikely driven by excess NADH relative to NAD+, but possibly due to carbon surplus in the TCA cycle.

Glycolysis and glutaminolysis both feed the TCA cycle and can produce lactate as a by-product (Smith et al, 2016). Next, we sought to determine whether the increased lactate production relative to pyruvate in the cells of Cluster 4 compared to Cluster 3 cells was due to greater consumption of glucose and/or glutamine. As expected, cells in Cluster 4 produced more lactate compared to those in Cluster 3 (unpaired student's t-test, $p = 0.006$; Fig. 3F), with some cell line-specific differences observed (Fig. EV2B). Likewise, there were cell line-specific differences in glucose and glutamine consumption over 24 h (Fig. EV2B), but somewhat surprisingly, the net consumption of glucose (unpaired student's t-test, $p = 0.02$) and glutamine (unpaired student's t-test, $p = 0.04$) were lower in Cluster 4 cells (Fig. 3F). In isolation, the higher lactate-to-glucose yield seen in Cluster 4 could be interpreted as higher aerobic glycolysis, whereas in Cluster 3 cells, proportionally more glucose was oxidized instead of being converted to lactate.

We postulated that Cluster 4 cells are more oxidatively competent and thus have surplus carbon that spills into lactate. To test this, we quantified the oxygen consumption rate as a readout of oxidative phosphorylation (OXPHOS) activity and, thus, TCA cycle fluxes. In line with our prediction, cells in Cluster 4 possessed greater OXPHOS activity than Cluster 3 cells (unpaired student's t-test, $p = 0.04$; Figs. 3G and EV3C). The more efficient respiration may correlate with our observation of more abundant TCA cycle metabolites relative to pyruvate (Fig. 3C) and suggests that the increase in lactate production (Fig. 3F) is a consequence of cells using pyruvate as a redox sink to regenerate NAD+.

To further resolve the intersection of glucose and glutamine metabolism at pyruvate, which underpins the formation of Clusters 3 and 4, we performed [U-$^{13}$C]-glucose and [U-$^{13}$C]-glutamine tracing experiments in a subset of cells selected from Clusters 3 and 4. As expected, lactate was produced mainly from glucose (78–92% enrichment). However, there were no differences in lactate and pyruvate enrichment between Clusters 3 and 4 (Figs. 3H and EV2D), nor were there differences in the enrichment of TCA cycle metabolites from glucose (Fig. EV2E). Therefore, combined with our measures of glucose consumption and lactate production, the higher TCA metabolite to pyruvate ratios in Cluster 4 cells compared to Cluster 3 cannot be explained simply by differences in glucose contribution and metabolism.

As such, we turned our attention to quantifying glutamine metabolism, as it is a major TCA cycle carbon source (Spinelli and Haigis, 2018), using [U-$^{13}$C]-glutamine tracing. Our first observation was that more glutamine carbons were incorporated in lactate, via either phosphoenolpyruvate carboxykinase or malic enzymes (Mansouri et al, 2017; Montal et al, 2015), in Cluster 4 cells than in Cluster 3 (Figs. 3I and EV2F). This increased efflux of glutamine from the TCA cycle was also observed as greater enrichment of glutamine carbons into m + 3 fructose 1,6-bisphosphate, and, to a lesser extent, phosphoenolpyruvate (Figs. 3J and EV2G), intermediates of gluconeogenesis. Combined, these data show that cells in Cluster 4 possessed a more oxidative phenotype that compensates for increased aerobic glycolysis with glutamine cataplerosis and explains the increased lactate production and more abundant TCA cycle metabolites in cells of Cluster 4, compared to Cluster 3 that had greater glucose and glutamine consumption and glucose oxidation (Fig. 3K). Furthermore, these new insights into glucose and glutamine metabolism and the discovery that some cells produce more lactate despite lower glucose consumption support the new insights into high flux metabolic pathways from our pathway-centric ratio-based analyses.

## Differences in TCA cycle and glutamine metabolism between Clusters 3 and 4 correlate with sensitivity to loss-of function

We further validated the outcomes of our pathway-centric metabolite ratio analysis of targeted metabolomic data (Fig. 2) and functional studies (Fig. 3) using in silico assessment of publicly available datasets. Specifically, we used the pan-cancer DEMETER (Tsherniak et al, 2017, Data ref: https://depmap.org/portal/data_page/?tab=allData) and Project Score (Behan et al, 2019, Data ref: https://score.depmap.sanger.ac.uk/downloads) loss-of-function

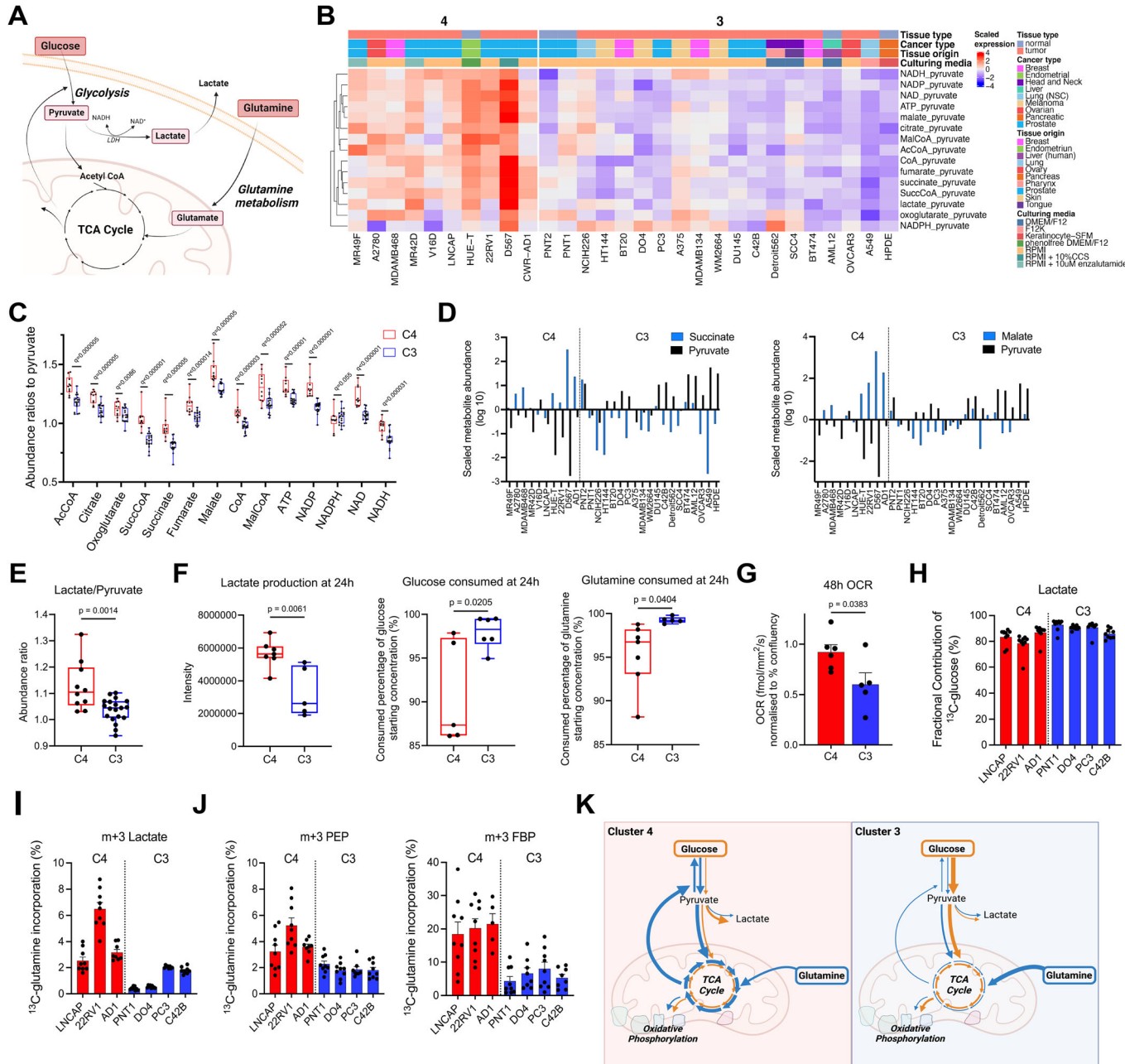

**Figure 3. Metabolite ratio-based signatures identifies differences in glutamine metabolism between clusters.**

(A) Schematic of glucose and glutamine sources to the TCA cycle. (B) Scaled metabolite ratio heatmap of Clusters 4 and 3, derived from TCA cycle pathway metabolites calculated using the precursor pyruvate as denominator. (C) Abundance ratios of TCA cycle metabolites to pyruvate (C4 $n = 10$ cell lines, C3 $n = 19$ cell lines. The center of each box plot represents the median, the box boundaries correspond to the upper and lower quartiles, and the whiskers extend to the minimum and maximal values). (D) Waterfall plots comparing the scaled metabolite abundance (log 10) of succinate and pyruvate (left), and malate and pyruvate (right) for cells of Clusters 3 and 4. (E) Abundance ratio of lactate to pyruvate (C4 $n = 10$ cell lines, C3 $n = 19$ cell lines. The center of each box plot represents the median, the box boundaries correspond to the upper and lower quartiles, and the whiskers extend to the minimum and maximal values). (F) Lactate production intensity and glucose and glutamine consumption (consumed % of starting substrate concentrations) after 24 h (C4 $n = 8$ cell lines, C3 $n$-6 cell lines. The center of each box plot represents the median, the box boundaries correspond to the upper and lower quartiles, and the whiskers extend to the minimum and maximal values). (G) Oxygen consumption rates (OCR) at 48 h, normalized to percent confluency (C4 $n = 6$, C3 $n$-5, 3 biological replicates and 4 technical replicates per cell line, mean ± standard error of the mean). (H) Fractional contribution of [U-13C]-glucose to intracellular lactate (C4 $n = 3$, C3 $n = 4$, 3 biological replicates and 3 technical replicates per cell line, mean ± standard error of the mean). (I) Fractional contribution of [U-13C]-glutamine to intracellular lactate (C4 $n = 3$, C3 $n = 4$, 3 biological replicates and 3 technical replicates per cell line, mean ± standard error of the mean). (J) Fractional contribution of [U-13C]-glutamine to m + 3 fructose 1,6-bisphosphate (FBP), and m + 3 phosphoenolpyruvate (PEP) (C4 $n = 3$, C3 $n = 4$, 3 biological replicates and 3 technical replicates per cell line, mean ± standard error of the mean). (K) Schematic of the C4 phenotype (increased glutamine to the TCA cycle, gluconeogenesis, and aerobic glycolysis) compared to C3 (increased glucose and glutamine consumption and glucose oxidation). Data information: (C) q (adjusted $p$) value vs Cluster 4 by multiple unpaired t-tests corrected for FDR, (D–G) $p$ value vs Cluster 4 by Unpaired student's t-test. Source data are available online for this figure.

screens, and the drug sensitivity PRISM (Corsello et al, 2020, Data ref: https://ndownloader.figshare.com/files/20237739) and GDSC2 (Yang et al, 2012, Data ref: https://cog.sanger.ac.uk/cancerrxgene/GDSC_release8.5/GDSC2_fitted_dose_response_27Oct23.xlsx) datasets and selected for gene or drug targets within all KEGG metabolic pathways (Fig. 4A). We then cross-referenced these large-scale cell line panels with cells that were members of Clusters 3 and 4 (Table EV1) and focused our analyses to test the hypothesis that cells in Cluster 4 were more susceptible to depletion of genes associated with OXPHOS, glutamine metabolism compared to Cluster 3 (Fig. 4B). In line with our hypothesis, Cluster 4 cells had greater sensitivity to genetic knockout of OXPHOS complexes I–V and glutaminase isoforms (Multiple unpaired student's t-tests corrected for FDR, adjusted $p < 0.05$; Fig. 4C). We also observed a trend for greater sensitivity to deletion of succinate dehydrogenase (SDHD; unpaired student's t-test, $p = 0.08$) in cells in Cluster 4 (Fig. EV3A). We complemented our targeted assessment of these datasets by determining the top 20 metabolic targets from all KEGG pathways that possessed the greatest difference between Clusters 4 and 3. From this unbiased approach, we identified members of OXPHOS, glutamine metabolism, and TCA cycle pathways in this list (highlighted in yellow in Fig. 4D,E). We also determined the top 20 list of the most different targets between Cluster 4 and 3 when we narrowed the coverage to just central carbon metabolism from all KEGG pathways (Fig. EV3B). Consistent with our other observations, the list of most sensitive targets again was enriched with enzymes from OXPHOS and TCA cycle pathways (Fig. EV3B). Finally, we developed a scoring system to consolidate the top 20 most sensitive pathways in Cluster 4 from all databases, and again found that the TCA cycle, OXPHOS, and glutamine metabolism were most vulnerable to genetic and drug targeting compared to Cluster 3 (Fig. EV3C). Combined, we have demonstrated that analyzing metabolomic data with a pathway-centric basis by using ratios identifies distinctive metabolic profiles that are evident in functional measures and loss-of-function screens.

### Pathway-centric ratio analyses of published datasets identify distinctive metabolic features in clusters of cancer cells

Finally, we set out to test if our pathway-centric approach can be applied to published metabolomic data and thereby support the major observations from our analyses. To achieve this, we selected Shorthouse et al (2022) (Data ref: https://massive.ucsd.edu/ProteoSAFe/dataset.jsp?accession=MSV000087155), which used primary data published in Cherkaoui et al (2022), as they expanded the metabolite coverage and introduced additional normalization and filtering. Between Shorthouse (179 cell lines, 1099 metabolites) and ours (57 cell lines, 50 metabolites), 39 metabolites and only 15 cell lines overlap. It is critical to highlight that the two metabolomics datasets are strikingly different. Firstly, Cherkaoui et al (2022) are untargeted data, reported as ion intensities, which was generated by flow injection (no LC separation, i.e., isomers like lysine and glutamine are not distinguished), whereas our targeted data, reported as concentrations from the included standards for all 50 metabolites, was generated by LC-MS/MS. Secondly, Cherkaoui et al (2022) adapted all cell lines to RPMI + 10% FCS, whereas we used normal/recommended media for each cell line (Table EV1).

Finally, there are differences in metabolite extraction protocol and recovery efficiency: Cherkaoui used monophasic 2:2:1 acetonitrile:methanol:water, whereas we used biphasic 1:1:2 methanol:water:chloroform. Next, we log10 transformed the Shorthouse data. Notably, 16 of 39 common metabolites had negative correlation coefficients (Fig. 5a). We would expect more positive correlations between datasets if metabolite profiles were cell line specific. This outcome is most likely explained by differences in extraction protocols and cell culture conditions, since extrinsic factors have nontrivial impacts on the metabolite profile (Golikov et al, 2022).

Despite this significant discrepancy, we continued and performed hierarchical clustering on the abundances of the 39 common metabolites from Shorthouse et al (2022). The combination of K-means clustering, gap statistics, elbow method and silhouette method determined that the optimal number of clusters of cells was five (Fig. EV4A). From this analysis, we identified that amino acids were more abundant in cells in Clusters 4 and 5 of Shorthouse et al (2022) than in the other Clusters (Fig. 5B), which we did not observe in our data (Fig. 1), which could arise from the use of a common culture media in Shorthouse et al (2022). Nonetheless, and consistent with our initial observations (Fig. 1), cell lines did not cluster according to cancer type, but these clusters consisted of different combinations of cell lines (Fig. 5B) compared to our analyses (Fig. 1). Additionally, there was no apparent organization of metabolites into pathways (Fig. 5B), which aligns with our initial results (Fig. 1). Overall, both datasets clearly show that metabolites do not cluster by pathway using standard cluster analyses.

We next calculated pathway-specific ratios using the Shorthouse et al (2022) dataset and determined that there were five optimum clusters using gap statistics, elbow method and silhouette method, consistent with the results generated using our metabolite ratios (Fig. EV4B). There were distinct metabolite patterns among ratios expressed relative to proline and glutamine, but not to pyruvate (Fig. 5C). Further, members of the clusters in the Shorthouse et al (2022) dataset (Fig. 5C) differed from ours, although there were some overlaps (Fig. 2; Table EV2). The cells in our Cluster 3 span Cluster 1 and 4 in the Shorthouse data, and our Cluster 2 and 5 cells span Cluster 3 and 5 in Shorthouse (Table EV2). This lack of consistency between the analysis of the Shorthouse dataset and ours is likely explained by the negative relationships in metabolite levels between our dataset and that of Shorthouse et al (2022) (Fig. 5A). Overall, these analyses of publicly available metabolite data support our conclusions that more meaningful differences in overall metabolic pathway function are identified using pathway-centric ratios compared to metabolite abundance alone.

## Discussion

Cell metabolism is dynamic and differs depending on context. From the early observations of Warburg (Warburg, 1925; Warburg and Minami, 1923) and the Coris (Cori and Cori, 1925), it is now well accepted that cancer cells metabolize glucose differently from non-tumor tissue. These differences are not a consequence of rewiring or transformation of the cascades of biochemical reactions that form glycolysis and PPP but essentially from increased uptake of extracellular glucose driving increased flux and production of end products, including lactate (DeBerardinis and Chandel, 2020).

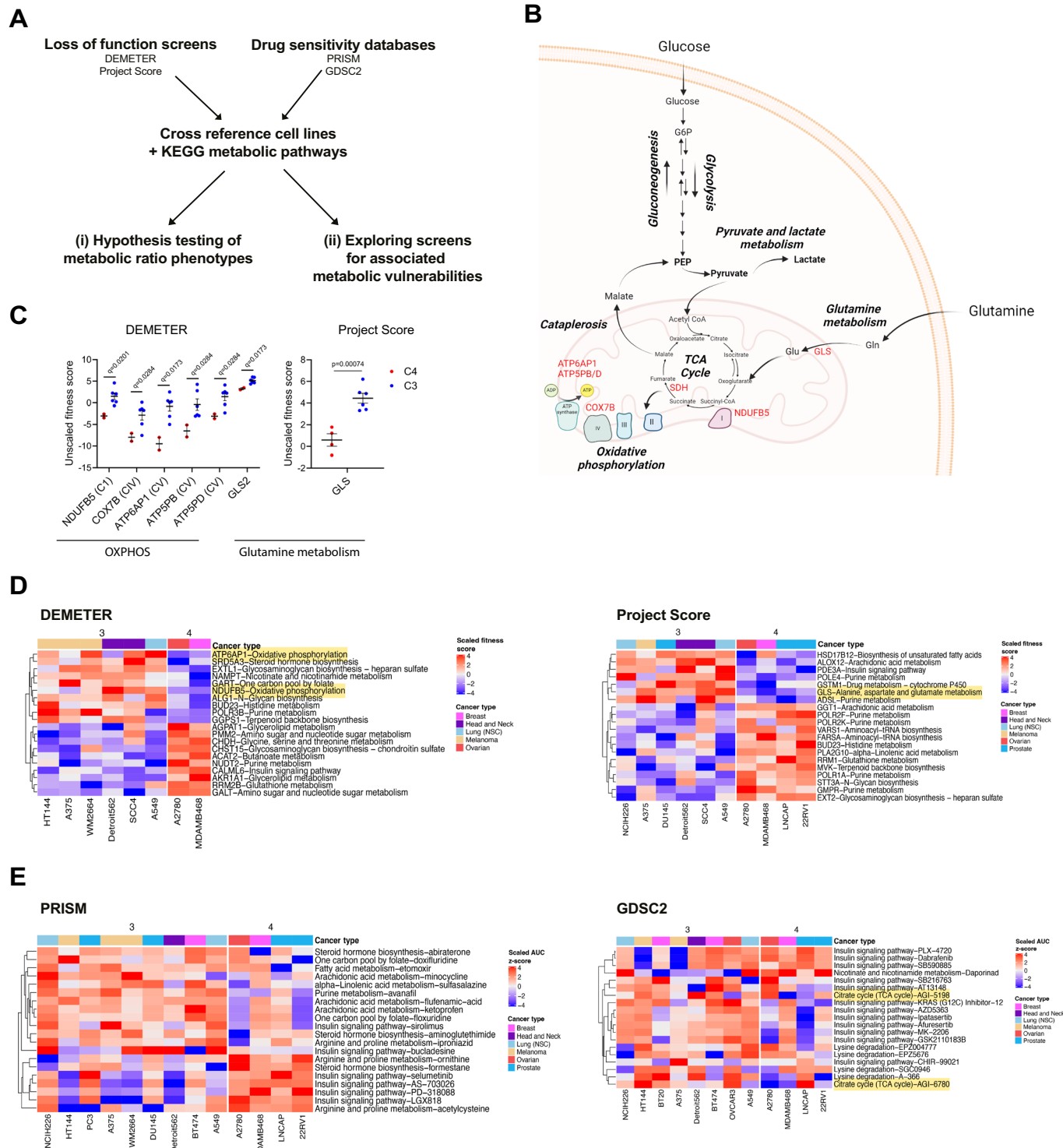

**Figure 4. Validating pathway-centric metabolite ratio clusters and substrate dependencies using loss-of-function screens and drug sensitivity databases.**

(A) Schematic of in silico analysis of loss-of-function and drug sensitivity screens against KEGG metabolic pathways and identified metabolic ratio phenotypes. (B) Schematic illustrating the C4 phenotype of greater utilization of glucose and glutamine identified by cluster analyses using metabolite ratios. Genes named in red were hypothesized to have greater sensitivity to gene knockouts or inhibition in C4 compared to C3. (C) Genes related to glucose and glutamine utilization phenotype of C4 with the greatest sensitivity to gene knockouts. (q (adjusted p) value vs Cluster 4 by multiple unpaired t-tests corrected for FDR, mean ± standard error of the mean, n = 2–6 cell lines). (D) Top 20 gene knockouts and associated KEGG pathways with the greatest fitness scores between C4 and C3 in DEMETER and Project Score databases. Genes and pathways matching C4 vulnerability model are highlighted in yellow. (E) Top 20 drugs and associated KEGG pathways with the greatest differential AUC z-scores between C4 and C3 in PRISM and GDSC2 databases. Drugs and pathways matching C4 vulnerability model are highlighted in yellow. Source data are available online for this figure.

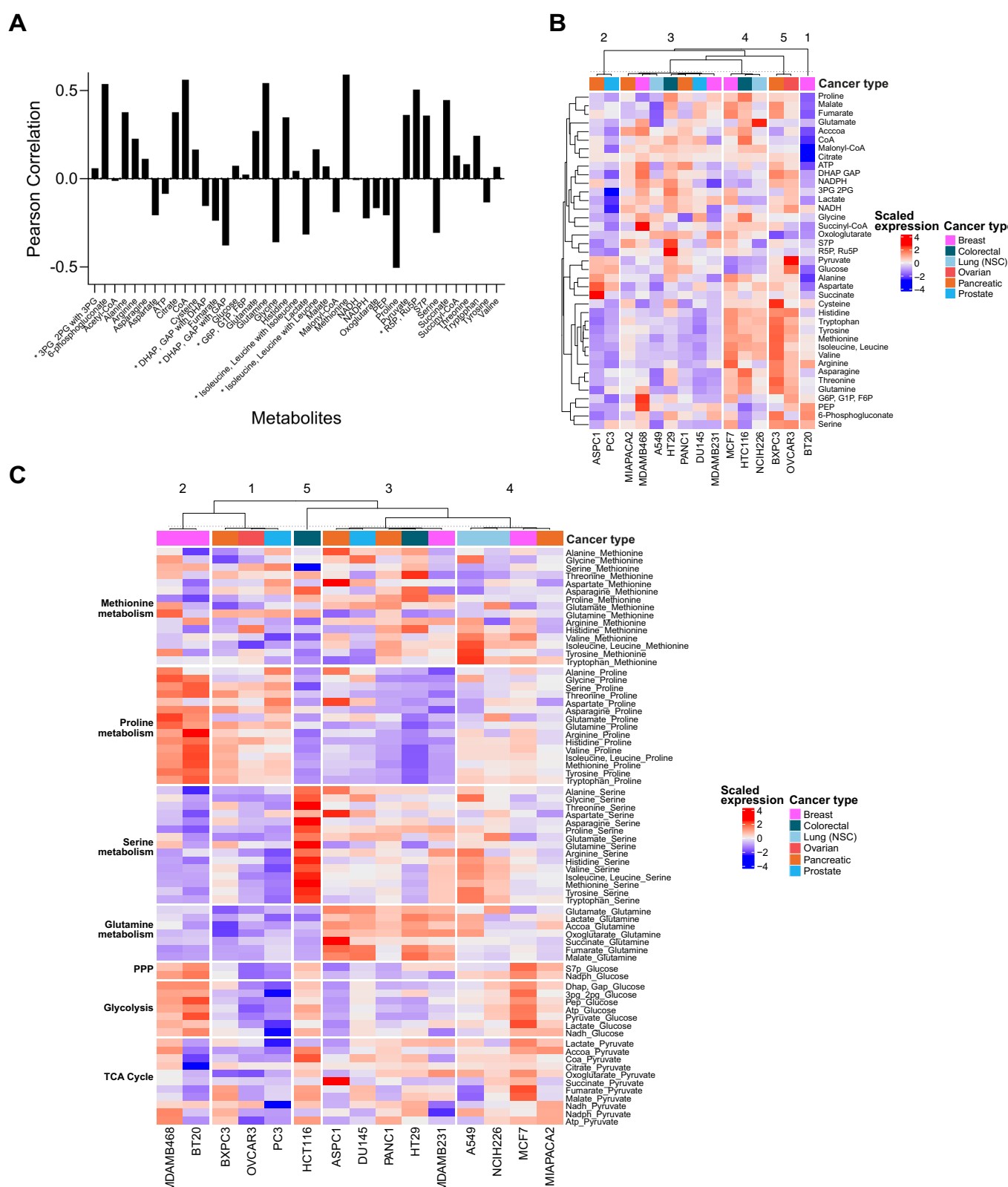

Alongside these well-documented changes in glucose metabolism, there have been significant advances in the understanding of the interaction between metabolic pathways (Sung et al, 2023), outside the well-established convergence of glucose, glutamine and fatty acid metabolism in the TCA cycle. A major challenge remains how to infer mechanistic differences in metabolism based on metabolomics data alone, since the latter may not correlate with pathway activity nor flux. Conventional dimensional reduction, clustering, and over-presentation methodologies rely on coordinated changes to infer co-dependency or co-regulation (Amara et al, 2022; Huang

◀

**Figure 5.   The metabolome landscape and pathway-centric metabolite ratios using published data.**

(A) Pearson correlation of metabolite concentrations in our dataset compared to metabolite ion intensities reported in Shorthouse et al (2022). * indicates decoupled metabolite isomers. (B) Heatmap of scaled metabolite expression across central carbon and amino acid metabolism within the common cell lines to our results and Shorthouse et al (2022) (n = 15). Cell lines color-coded by cancer type and tissue type. (C) Heatmap of scaled metabolite ratios by pathways covered in the untargeted metabolomics approach by Shorthouse et al (2022). Ratios are calculated for all metabolites of a specific pathway against a precursor metabolite. Source data are available online for this figure.

and Wang, 2022), but the effectiveness of subsequent interpretations may be hinge on how the literature has delineated metabolite memberships (Mahajan et al, 2024), which are continuously honed over time.

Our first round of clustering, which was based on metabolite concentrations of proliferating cells, formed groups that contained both epithelial and tumor-derived, and dismissed culturing conditions, tissue type and origin, and mutation status of oncogenic drivers as potential factors influencing cell metabolism. However, this approach failed to generate any metabolic subtype signatures or hypotheses centered on pathway activity differences that could be functionally evaluated. Consequently, we took a physiologically-based approach and transformed our metabolomic data into pathway-centric ratios and explored the relationships between reactant and product metabolites of central carbon and amino acid pathways. The approach draws upon thermodynamic principles in terms of reaction equilibrium and Gibbs energy (Park et al, 2016), and in practice, there is robust evidence for coordinated changes among proximal and hub metabolites (Martínez-Reyes and Chandel, 2020). Using metabolite ratios, we identified five clusters of cell lines, two of which (Cluster 3 and 4) we surmised to differ in TCA cycle activity based on the levels of TCA cycle metabolites relative to pyruvate. Indeed, the ensuing functional assay of glucose and glutamine metabolism verified pathway activity differences between Cluster 3 and 4, with Cluster 4's elevated TCA cycle metabolites showing concordance with higher lactate production and the shift from glycolysis to an increased glutamine oxidation and cataplerosis. This work highlights the value of pathway-centric ratio-based data transformation in distinguishing metabolic pathway activity in a pan-cancer cell line panel and, therefore, supports the concept of a physiological-based approach to analyze metabolomic data.

Many tumors are reliant on glutamine as a critical carbon and nitrogen source, with glutaminase inhibition shown to be an effective therapy in several cancer types, such as triple-negative breast cancer (Gross et al, 2014), non-small cell lung cancer (van den Heuvel et al, 2012), and head and neck cancer (Wicker et al, 2021). Our findings showed that Cluster 4 cells had higher OXPHOS levels yet consumed less glucose and glutamine; these cells appear more carbon efficient. The relatively more plentiful TCA cycle metabolites may sustain higher TCA cycle fluxes, which are tightly coupled to OXPHOS, and buffer critical anabolic and signaling functions (Martínez-Reyes and Chandel, 2020). Glutamine's proximity means oxidizing glutamine repletes TCA cycle metabolites more directly than glucose (Quek et al, 2022), but the displacement of glucose oxidation further entrenched the aerobic glycolysis phenotype as seen in Cluster 4 cells. Additionally, glutamine cataplerosis and the contribution of PEP carboxykinase to gluconeogenesis, converting oxaloacetate to PEP, augment the supply of biomass precursors, which have been documented in several cancer types, including liver (Liu et al, 2018) and lung

(Vincent et al, 2015). For example, glutamine-derived lactate production via glutaminolysis in glioblastoma cells helps produce NADPH and support fatty acid synthesis (DeBerardinis et al, 2007). Overall, we speculate that the increased glutamine utilization among Cluster 4 cells may confer greater fitness to support proliferation and survival.

Excitingly, our assertion of elevated OXPHOS and enhanced glutamine utilization in Cluster 4 matched data mining results derived from drug sensitivity (Corsello et al, 2020; Yang et al, 2012) and loss-of-function screens (Behan et al, 2019; Tsherniak et al, 2017). Among the top 20 loss-of-function screens, Cluster 4 cells were most sensitive to targeting the TCA cycle, OXPHOS, glycolysis, and glutamate metabolism pathways, compared to Cluster 3 cells. Namely, we found Cluster 4 cells significantly more sensitive to glutaminase gene knockout and inhibitors. These findings highlight the potential of a physiological pathway-centric approach to translating metabolite signatures into effective strategies for identifying druggable vulnerabilities in glucose and glutamine metabolism.

A key limitation of our approach is coverage. Our targeted LC-MS method covers central carbon metabolism, and thus we have focused on the TCA cycle as it is a convergent point for glucose and glutamine utilization; however, whether these relationships are consistent for other TCA cycle substrates, such as branched-chain amino acids and fatty acids (Neinast et al, 2019; Schoors et al, 2015) remains to be determined. Another major lesson from our approach was that the most insightful outcome of our clustering analysis came from using a hub metabolite (e.g., pyruvate) as the denominator rather than a starting substrate (i.e., glucose, glutamine, serine). Since we only included one hub metabolite in our pathway cluster analysis, it is conceivable that investigating other metabolic hubs not covered in our targeted approach, such as $NAD^+$ (Benedetti et al, 2023), could identify other distinctive signatures and targetable vulnerabilities. Perhaps the abundance of hub metabolites, where multiple pathways converge, are less prone to isolated variation or are tightly regulated and thus more effective at distinguishing metabolic changes in a ratio approach. For example, pairing glutamate, a transamination hub metabolite, with 2-hydroxygltuarate was effective at predicting IDH mutation status in patients with glioma (Hua et al, 2024). A potential issue with our pathway-centric ratio approach—common to all 'omics data analyses using conventional dimensional reduction, clustering, and over-representation methods—is that transforming data through the use of a denominator metabolite (in our case) or normalization factor risk introducing bias, particularly due to large measurement variability and incorrect error propagation. For example, in our experiments the %RSD of raw pyruvate measurements was 11% for the quality control samples (pool, 10 injections), 74% for all 78 samples, and an average of 14% for the technical replicates (26 cell lines, n = 3 each). Overall, this means the error associated with pyruvate measurements was generally small relative

to the observed differences between cell lines. It is possible that using other metabolites that have greater variability as the denominator will significantly influence the outcomes of these types of analyses. Finally, like all cell culture-based studies, there are questions about the physiological relevance and translation of our results to in vivo settings. Many studies have highlighted major issues with traditional cell culture. For example, glutamine anaplerosis and glutaminase dependence in cultured cancer cells is, in part, due to limited cystine availability in standard cell culture media, and also explains why the same cells rely less on glutamine catabolism to proliferate in vivo (Davidson et al, 2016; Muir et al, 2017). Others have reported that medium composition (traditional versus human plasma-like medium) impacts gene essentiality in CRISPR-based screens using human cancer cells (Rossiter et al, 2021), yet, rapid depletion of nutrients in physiological media (e.g., from 25 mM glucose in DMEM to 5.5 mM in Plasmax) leads to starvation responses, apoptotic signaling, and endoplasmic reticulum stress gene expression signatures (Gardner et al, 2022). Nonetheless, we believe that our approach of transforming metabolomic data into pathway-centric ratios can be applied to datasets generated from clinical tumor tissue and pre-clinical cancer models.

A major motivation for the current study was to identify common metabolic signatures that arise from various genomic bases that can form the foundation for a simpler therapeutic approach across cancer types. The outcomes of this study have, in part, provided evidence that this may be achievable. Our results highlight the potential of using physiologically-based, pathway-centric metabolite ratios to gain insights into the convergent or recurrent pathway mechanisms within subsets of diverse cancer types and identify targetable vulnerabilities. Combined with existing large-scale metabolomic datasets, our approach may lay the foundation to accelerate the ongoing efforts to profile cancer metabolism for future therapeutic advances and repurposing metabolic targeting-based therapeutics in a pan-cancer setting.

# Methods

### Reagents and tools table

| Reagent/Resource | Reference or Source | Identifier or Catalog Number |
|---|---|---|
| **Experimental models** | | |
| MCF10A | ATCC | ATCC CRL-10317 RRID: CVCL_0598 |
| MDA-MB-231 | ATCC | ATCC HTB-26 RRID: CVCL_0062 |
| BT474 | ATCC | ATCC HTB-20 RRID: CVCL_0179 |
| MDA-MB-468 | ATCC | ATCC HTB-132 |
| MCF7 | ATCC | ATCC HTB-22 |
| BT20 | ATCC | ATCC HTB-19 |
| MDA-MB-134 | ATCC | ATCC HTB-23 |
| PNT1 | ECACC | ECACC 95012614 RRID: CVCL_2163 |
| PNT2 | Prof. Lisa Butler, SAHMRI | (Nassar et al, 2020) |

| Reagent/Resource | Reference or Source | Identifier or Catalog Number |
|---|---|---|
| C4-2B | ATCC | ATCC CRL-3315 |
| PC-3 | ATCC | ATCC CRL-1435 |
| 22Rv1 | ATCC | ATCC CRL-2505 RRID: CVCL_1045 |
| LNCaP | ATCC | ATCC CRL-1740 RRID: CVCL_1379 |
| DU145 | A/Prof. Luke Selth, Flinders University | ATCC HTB-81 |
| CWR-AD1 | A/Prof. Luke Selth, Flinders University | (Li et al, 2013; Nyquist et al, 2013) |
| CWR-D567 | A/Prof. Luke Selth, Flinders University | (Nyquist et al, 2013) |
| VCAP | Prof. Lisa Butler, SAHMRI | (Nassar et al, 2020) |
| V16D | Prof. Lisa Butler, SAHMRI | (Nassar et al, 2020) |
| MR49F | Prof. Lisa Butler, SAHMRI | (Bishop et al, 2017) |
| MR42D | Prof. Lisa Butler, SAHMRI | (Bishop et al, 2017) |
| AML12 | Prof. Rob Parton, University of Queensland | (Nagarajan et al, 2019) |
| PH5CH8 | A/Prof. Susan McLennan, University of Sydney | (Nagarajan et al, 2019) |
| HEPG2 | A/Prof. Susan McLennan, University of Sydney | (Nagarajan et al, 2019) |
| HUH7 | Prof. Mark Gorrell, University of Sydney | RRID: CVCL_0336 |
| SKHEP1 | Prof. Mark Gorrell, University of Sydney | RRID: CVCL_0525 |
| HEPA1-6 | Prof. Mark Gorrell, University of Sydney | RRID: CVCL_0327 |
| HUE-T | Dr Frances Byrne, UNSW | (Byrne et al, 2014) |
| MAD11 | Dr Frances Byrne, UNSW | (Byrne et al, 2014) |
| Ishikawa | Prof. Kyle Hoehn, UNSW | RRID: CVCL_2529 |
| MFE296 | Prof. Kyle Hoehn, UNSW | RRID: CVCL_1406 |
| MFE319 | Prof. Kyle Hoehn, UNSW | RRID: CVCL_2112 |
| AN3CA | Prof. Kyle Hoehn, UNSW | RRID: CVCL_0028 |
| RL952 | Prof. Kyle Hoehn, UNSW | RRID: CVCL_0505 |
| KLE | Prof. Kyle Hoehn, UNSW | RRID: CVCL_1329 |
| HEC1A | Prof. Kyle Hoehn, UNSW | RRID: CVCL_0293 |
| U251 | Prof. Lenka Munoz, University of Sydney | RRID: CVCL_0021 |
| U87 | Prof. Lenka Munoz, University of Sydney | RRID: CVCL_3429 |
| HPDE | A/Prof Thomas Grewal, University of Sydney | RRID: CVCL_4376 |
| PANC1 | A/Prof Thomas Grewal, University of Sydney | RRID: CVCL_0480 |
| MIAPACA-2 | A/Prof Thomas Grewal, University of Sydney | RRID: CVCL_0428 |
| BXPC3 | A/Prof Thomas Grewal, University of Sydney | RRID: CVCL_0186 |
| ASPC1 | A/Prof Thomas Grewal, University of Sydney | RRID: CVCL_0152 |

| Reagent/ Resource | Reference or Source | Identifier or Catalog Number |
|---|---|---|
| A2780 | A/Prof. David Croucher, Garvan Institute | RRID: CVCL_0134 |
| OVCAR3 | A/Prof. David Croucher, Garvan Institute | RRID: CVCL_0465 |
| SKOV3 | A/Prof. David Croucher, Garvan Institute | RRID: CVCL_0532 |
| A375 | Dr. Lorey Smith, Peter MacCallum Cancer Centre | ATCC CRL-1619 RRID: CVCL_0132 |
| HT144 | Dr. Lorey Smith, Peter MacCallum Cancer Centre | ATCC HTB-63 RRID: CVCL_0318 |
| WM266.4 | Dr. Lorey Smith, Peter MacCallum Cancer Centre | ATCC CRL-1676 RRID: CVCL_2765 |
| DO4-M1 | Dr. Lorey Smith, Peter MacCallum Cancer Centre | (Parmenter et al, 2014) |
| A549 | Dr. Lorey Smith, Peter MacCallum Cancer Centre | (Chen et al, 2019) |
| NCI-H226 | Dr. Lorey Smith, Peter MacCallum Cancer Centre | RRID: CVCL_1544 |
| Detroit562 | A/Prof Thomas Grewal, University of Sydney | RRID: CVCL_1171 |
| SCC4 | A/Prof Thomas Grewal, University of Sydney | RRID: CVCL_1684 |
| HT29 | A/Prof. Kellie Charles, University of Sydney | RRID: CVCL_A8EZ |
| HCT116 | A/Prof. Kellie Charles, University of Sydney | RRID: CVCL_0291 |
| SW-48 | A/Prof. Kellie Charles, University of Sydney | RRID: CVCL_1724 |
| SW-1417 | A/Prof. Kellie Charles, University of Sydney | RRID: CVCL_1717 |
| **Chemicals, Enzymes, and other reagents** | | |
| Enzalutamide MDV3100 | Selleckchem | Cat#S1250 |
| D-Glucose | Sigma-Aldrich | Cat#273053 |
| L-Glutamine | Gibco | Cat#25030081 |
| Fatty acid free BSA | Bovogen | Cat#BSAS0.10 |
| D-Glucose (U-13C6, 99%) | Cambridge Isotope Laboratories, Inc. | Cat#CLM-1396 |
| L-Glutamine (U-13C5, 99%) | Cambridge Isotope Laboratories, Inc. | Cat#CLM-1822-H |
| **Software** | | |
| GraphPad Prism V9 | GraphPad Software | https://www.graphpad.com/ |
| MATLAB | Mathworks | https://au.mathworks.com/products/matlab.html |
| MSConvert | N/A | (Chambers et al, 2012) |
| MSDial | Riken | http://prime.psc.riken.jp/compms/msdial/main.html |
| Biorender | Biorender | https://biorender.com/ |
| **Other** | | |
| Resipher | Lucid Scientific | https://lucidsci.com/ |
| Incucyte-SX5 | Sartorius | https://www.sartorius.com/ |

## Cell lines and culture conditions

A total of 57 adherent cell lines (8 normal and 49 tumor) spanning 11 cancer types were used in this study. Cell lines obtained from vendors or that were kindly provided by academic labs (Table EV1). Mutational status of common oncogenes was characterized by cross-referencing the Cellosaurus database, COSMIC database, and Depmap portal. Standard culture conditions for each cell line were used to avoid the associated stress and change in cell population associated with switching culture media. This included the enzalutamide-resistant prostate cancer cells MRF49F and MR42D requirement to be cultured in enzalutamide, and CWR-D567 prostate cancer cells that have been engineered to express an androgen receptor variant that lacks the ligand-binding domain and so grown in androgen-deplete charcoal-stripped serum. In general, culturing media were supplemented with 10% fetal bovine serum (Cytiva Hyclone) and 1% penicillin/streptomycin (Gibco) unless stated otherwise (Table EV1). Cells were incubated at 37 °C and 5% $CO_2$. The full panel of cells were generated across 6 batches over a period of 3 years due to COVID restrictions and included overlapping cell lines to correct for batch effects. Cell lines were validated periodically by Garvan Molecular Genetics using a test based on the Powerplex 18D kit (DC1808, Promega) and tested for mycoplasma every 3 months (MycoAlert™ mycoplasma detection kit, Lonza).

## Metabolomics experiments

Cells were seeded in triplicate in 6-well plates at a density of $5 \times 10^5$ cells/well in 2 mL of media. After 24 h, the media was removed, and cells were washed once with 2 mL ice-cold 0.9% w/v NaCl. Cells were then scraped with 300 μL of extraction buffer, EB (1:1 LC/MS methanol:water (Optima) + 0.1x internal standards comprised of non-endogenous polar metabolites (2-morpholinoethanesulfonic acid, D-camphor-10-sulfonic acid, and deuterated thymidine) and transferred to a 1.5 mL microcentrifuge tube. A further 300 μL of EB was added to the well and combined in the tube. 600 μL chloroform (Honeywell) was added before vortexing and incubating on ice for 10 min. Tubes were vortexed briefly and centrifuged at $15,000 \times g$ for 10 min at 4 °C. The aqueous layer was collected and dried without heat, using a Savant SpeedVac (Thermo Fisher). Dried samples were resuspended in 40 mL Amide buffer A (20 mM ammonium acetate, 20 mM ammonium hydroxide, 95:5 HPLC $H_2O$:Acetonitrile (v/v)) and vortexed and centrifuged at $15,000 \times g$ for 5 min at 4 °C. 20 μL of supernatant was transferred to HPLC vials containing 20 μL acetonitrile for LC-MS analysis of amino acids and glutamine metabolites. The remaining 20 μL of resuspended sample was transferred to HPLC vials containing 20 μL LC-MS $H_2O$ for LCMS analysis of glycolytic, pentose phosphate pathway, and TCA cycle metabolites. Amino acids and glutamine metabolites were measured using the Vanquish-TSQ Altis (Thermo) LC-MS/MS system. Analyte separation was achieved using a Poroshell 120 HILIC-Z Column (2.1 × 150 mm, 2.7 μm; Agilent) at ambient temperature. The pair of buffers used were Amide buffer A and 100% acetonitrile (Buffer B), flowed at 200 μL/ min; injection volume of 5 μL. Glycolytic, PPP and TCA cycle metabolites were measured using 1260 Infinity (Agilent)-QTRAP6500+ (AB Sciex) LC-MS/MS system. Analyte separation was achieved using a Synergi 2.5 μm Hydro-RP 100 A LC Column

(100 × 2 mm) at ambient temperature. The pair of buffers used were 95:5 (v/v) water:acetonitrile containing 10 mM tributylamine and 15 mM acetic acid (Buffer A) and 100% acetonitrile (Buffer B), flowed at 200 µL/min; injection volume of 5 µL. Raw data from both LC-MS/MS systems were extracted using MSConvert (Chambers et al, 2012) and in-house MATLAB scripts. Concentrations of metabolites were calculated against a standard curve of polar and amino acid metabolite standards similarly extracted as above. Log10 normalization was performed on the metabolite concentration data.

## Metabolomics batch correction

Cell line samples were quantified over 6 batches during the experiment including overlapping cell lines across each batch. To eliminate potential batch effects, we applied the normalization method: Removing Unwanted Variation-Ill (RUV-III) (Molania et al, 2019). This approach has been successfully used in multiple omics, including bulk RNA-seq, single-cell RNA as well as metabolite data. We used the RUV-III version as it can account for replicates in the estimation of unwanted noise and replicate samples are a key part of our experimental design. We set k, the number of unwanted factors, to 9. The cell lines that were measured across multiple runs were used as replicates for the batch correction algorithm and to assess the quality of the batch correction output. That is, after batch correction, we performed K-means clustering with Pearson correlation to ensure the same cell lines across multiple runs were clustered in the same group.

## Metabolomics clustering analysis

Clustering was performed on the batch-corrected matrix using K-means algorithms with "1 - Pearson correlation" as the distance matrix (Gu et al, 2016) implemented in the ComplexHeatmap package. The goodness of number of clusters was evaluated using three standard heuristics, namely gap statistics (Tibshirani et al, 2002), elbow method (Thorndike, 1953), and silhouette method (Rousseeuw, 1987), to achieve sufficient separation between clusters and avoid overfitting. We visualized the results as a heatmap using the ComplexHeatmap package. The clustering procedure was repeated 1000 times and the consensus was taken to ensure a more robust clustering result. The clustering strategy was applied to both the original batch-corrected matrix and the ratio-transformed matrix. We visually assess whether the obtained clustering was not influenced by potential confounding factors, such as culturing conditions and common oncogenic gene mutations and tissue type. This is achieved by adding a color-coded legend on culturing conditions and mutational information to the clustering heatmap and demonstrating that no patterns exist.

The original batch-corrected matrix was subset to target metabolites from several key metabolic pathways including the TCA cycle, pentose phosphate pathway, glycolysis, amino acid metabolism, and glutamine metabolism, followed by clustering analysis. For the ratio-transformed matrix, batch-corrected abundance ratios were calculated between a precursor metabolite for each pathway (used as the denominator) and the remaining pathway metabolites. The corresponding precursor (denominator) metabolites were pyruvate for the TCA cycle, glucose for pentose phosphate pathway and glycolysis, and glutamine for the glutamine

metabolism pathway. For amino acid pathways, serine, proline, and methionine were used as the precursor metabolites. The clustering structure determined from the processes above were also used to visualize the oncogene mutation status, tissue type, cancer type, tissue origin, and culturing media type of the cell lines using a heatmap and inspect the relationship of these variables with the determined clusters.

## U-$^{13}$C stable isotope tracing

Cells were seeded in triplicate in 6-well plates at a density of $5 \times 10^5$ cells/well in 2 mL of media. After 24 h wells were washed with warm PBS and media replaced with 600 µL of DMEM no glucose, no glutamine media supplemented with 2% (wt/vol) FA-free BSA, 150 µM palmitate, 5 mM glucose and 1 mM glutamine, replaced with their respective U-$^{13}$C forms. Cells were incubated in U-$^{13}$C containing medium for 6 h. Samples were extracted and measured using the Vanquish-TSQ Altis and Agilent-QTRAP6500+.

## Extracellular substrate experiments

Cells were seeded in triplicate in 6-well plates at a density of $5 \times 10^5$ cells/well in 2 mL of media. After 24 h wells were washed with warm PBS and media replaced with 1 mL of DMEM no glucose, no glutamine media supplemented with 5 mM glucose, 1 mM glutamine and 150 µM palmitate. 100 µL of extracellular media were collected from wells at 3, 6, 12, 24 h timepoints. Media samples were centrifuged at $16,000 \times g$ for 5 min at 4 °C and the supernatant collected for subsequent extraction and LC-MS analysis.

To extract media samples for LC-MS, 20 µL of supernatant media was first diluted with 80 µL water and vortexed, and then 10 µL of the diluted media was transferred to 90 µL of extraction buffer containing 1:1 (v/v) acetonitrile and methanol + 1x internal standards (non-endogenous standards) at −30 °C. The mixture was centrifuged at $12,000 \times g$ for 5 min at 4 °C and transferred into HPLC vials for LC-MS analysis measured using the Vanquish-TSQ Altis.

## Oxygen consumption rate

Cells were seeded in 96-well plates (Nunc) in 100 µL basal medium. Oxygen consumption rates were continuously measured using Resipher (Lucid Scientific) at 37 °C, 10% $CO_2$ as per manufacturer instructions over 48 h, starting 24 h post-seeding. Media volumes were replenished at 24 h. Cell-free wells contained 200 µL of PBS to avoid evaporation. To account for differences in cell growth over 48 h, parallel plates were similarly cultured, and media replenished at 24 h for cell confluency measured using IncuCyte-SX5 (Sartorius).

## Analysis of drug and CRISPR databases

Drug sensitivity area under the curve (AUC) data were downloaded from the PRISM (Corsello et al, 2020, Data ref: https://ndownloader.figshare.com/files/20237739) and GDSC2 (Release 8.4, July 2022) (Yang et al, 2012, Data ref: https://cog.sanger.ac.uk/cancerrxgene/GDSC_release8.5/GDSC2_fitted_dose_response_27Oct23.xlsx) databases. Loss-of-function fitness score data were downloaded from the DEMETER (Tsherniak et al,

2017, Data ref: https://depmap.org/portal/data_page/?tab=allData) and Project Score (July 2021) (Behan et al, 2019, Data ref: https://score.depmap.sanger.ac.uk/downloads) databases. First, cell lines were filtered by overlapping cell lines within our panel, and then inhibitor or loss-of-function gene targets were filtered by KEGG metabolic pathway genes. We then performed a differential expression analysis on the drug response of the cell lines belonging to clusters of interest, specifically the response of Cluster 3 cell lines versus Cluster 4 cell lines. The top 20 drugs or gene targets with the greatest differential response between Clusters 3 and 4 was identified.

### Analysis of publicly available metabolomic data

Metabolomic data previously reported in Shorthouse et al (2022) (Data ref: https://massive.ucsd.edu/ProteoSAFe/dataset.jsp?accession=MSV000087155) was downloaded, filtered by overlapping cell lines within our panel, then overlapping metabolites, and then log10 transformed. The resulting dataset was analyzed using the same approaches described in Metabolomics clustering analysis.

### Statistical analysis for in vitro and database analysis

For all $^{13}C$-tracing, extracellular, and oxygen consumption experiments, at least 3 technical replicates and 3 independent biological replicates were used for each sample group. Descriptive data summary in Figs. 3 and 4 and Extended View Figs. 2 and 3 were presented as mean ± standard error of the mean (SEM), mean ± standard deviation (SD), or mean ± min to max values, as indicated in each figure legend. We determine statistically significant differences between cell clusters 3 and 4 in $^{13}C$-tracing, extracellular, and oxygen consumption experiments and loss-of-function analysis by performing multiple unpaired student's t-tests with false discovery rate multiple comparisons correction set at 5% using the Benjamini, Krieger, and Yekutieli FDR method and unpaired Student's t-test (implemented in Prism GraphPad V10). We assessed the differences between clusters 3 and 4 of oxidative phosphorylation gene loss-of-function by multiple unpaired student's t-tests with false discovery rate multiple comparisons correction set at 5% using the Benjamini, Krieger, and Yekutieli FDR method (implemented in Prism GraphPad V10). Outcomes from statistical testing is reported as exact $p$ value or $q$ value (adjusted $p$ value), with $p < 0.05$ interpreted as statistically significant different and $p > 0.05$ not statistically different otherwise. The R software and packages were used for clustering methods and heatmaps as reported in previous sections. Schematic diagrams were created with Biorender.com.

No blinding was performed in these studies.

## Data availability

The datasets and computer code produced in this study are available in the following databases: Metabolomic data: GitHub (https://github.com/lakeeeq/HoyLab-pancancer-clustering). Data used for in silico analyses are appropriately cited in the manuscript. All other data are available in the manuscript or the supplementary materials.

The source data of this paper are collected in the following database record: biostudies:S-SCDT-10_1038-S44320-025-00099-0.

## Peer review information

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

## Acknowledgements

NTS was supported by the Australian Rotary Health/Rotary Club of Blacktown City 'Mel Gray' PhD scholarship. AJH was supported by a Robinson Fellowship and funding from The University of Sydney. We thank the Sydney Mass Spectrometry facility for access to LC-MS instruments; and all collaborators who kindly donated cell lines.

## Author contributions

**Nancy T Santiappillai**: Formal analysis; Validation; Investigation; Visualization; Writing—original draft; Project administration; Writing—review and editing. **Yue Cao**: Software; Formal analysis; Validation; Visualization; Methodology; Writing—review and editing. **Mariam F Hakeem-Sanni**: Investigation; Writing—review and editing. **Jean Yang**: Supervision; Methodology; Writing—review and editing. **Lake-Ee Quek**: Conceptualization; Software; Formal analysis; Supervision; Methodology; Project administration; Writing—review and editing. **Andrew J Hoy**: Conceptualization; Resources; Formal analysis; Supervision; Funding acquisition; Project administration; Writing—review and editing.

Source data underlying figure panels in this paper may have individual authorship assigned. Where available, figure panel/source data authorship is listed in the following database record: biostudies:S-SCDT-10_1038-S44320-025-00099-0.

## Disclosure and competing interests statement

The authors declare no competing interests.

# Expanded View Figures

**Figure EV1.   Pathway-centric metabolite ratios of all targeted metabolic pathways.**

(**A**) Heatmap of scaled metabolite ratios by pathways covered in the targeted metabolomics approach. Ratios calculated for all metabolites of a specific pathway against a precursor metabolite. (**B**) Gap statistic, elbow method and silhouette method that determined the ideal number of clusters alongside K-means clustering based on Pearson's correlation coefficient method. (**C**) Clusters of cell lines from (**A**) appended with color coded legends for mutant status of common oncogenic drivers, culturing media conditions and tissue origins. Source data are available online for this figure.

▶

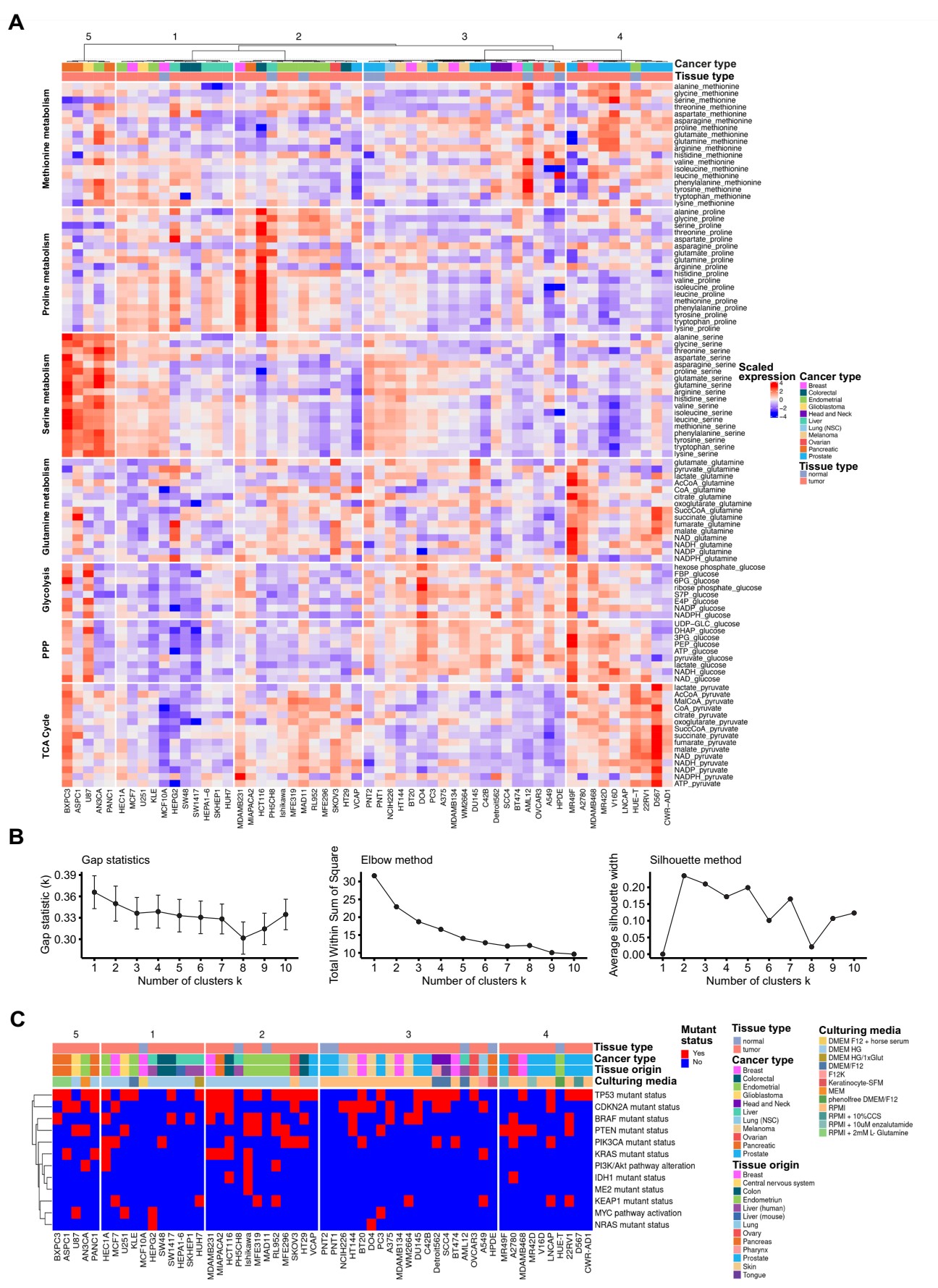

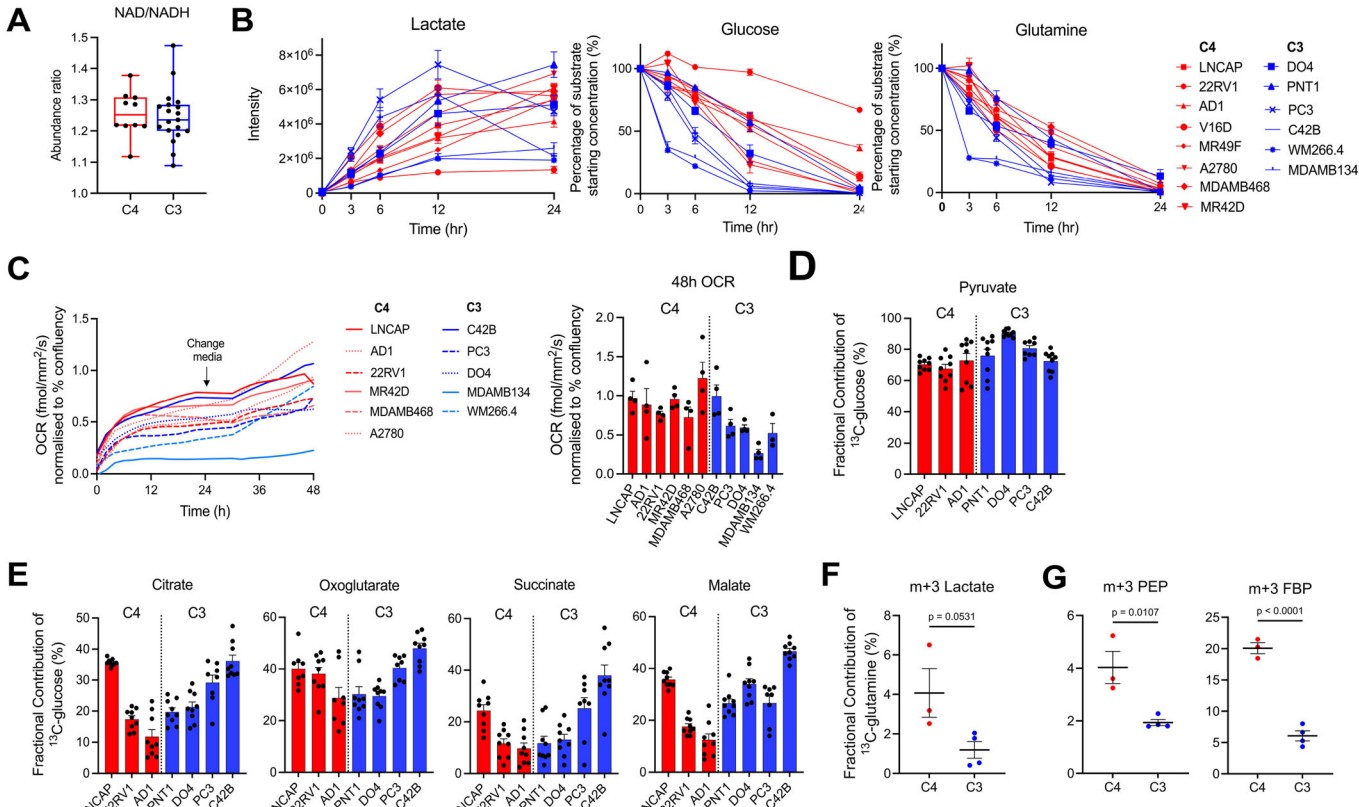

**Figure EV2. Substrate preferences for cell line clusters identified by TCA cycle ratios.**

(A) NAD+ to NADH ratio in cells of Clusters 4 and 3 (C4 $n = 10$ cell lines, C3 $n = 19$ cell lines, Unpaired student t-test, $p = 0.76$. The center of each box plot represents the median, the box boundaries correspond to the upper and lower quartiles, and the whiskers extend to the minimum and maximal values). (B) Lactate production and glucose and glutamine consumption over 24 h (C4 $n = 8$, C3 $n$-6, 3 biological replicates and 3 technical replicates per cell line, mean ± standard error of the mean). (C) Oxygen consumption rates (OCR) measured over 48 h, normalized to % confluency. Media changed at 24 h (left). OCR measurement for cell lines at 48 h (right) (C4 $n = 6$, C3 $n$-5, 3 biological replicates and 4 technical replicates per cell line). (D) Fractional contribution of [U-$^{13}$C]-glucose to intracellular pyruvate (C4 $n = 3$, C3 $n = 4$, 3 biological replicates and 3 technical replicates per cell line, mean ± standard error of the mean). (E) Fractional contribution of [U-$^{13}$C]-glucose to TCA metabolites citrate, oxoglutarate, succinate and malate (C4 $n = 3$, C3 $n = 4$, 3 biological replicates and 3 technical replicates per cell line, mean ± standard error of the mean). (F) Fractional contribution of [U-$^{13}$C]-glutamine to intracellular lactate (C4 $n = 3$, C3 $n = 4$, mean of 3 biological replicates and 3 technical replicates per cell line, $p$ value vs Cluster 4 by Unpaired student's t-tests, mean ± standard error of the mean). (G) Fractional contribution of [U-$^{13}$C]-glutamine to m + 3 fructose 1,6-bisphosphate (FBP), and m + 3 phosphoenolpyruvate (PEP) (C4 $n = 3$, C3 $n = 4$, mean of 3 biological replicates and 3 technical replicates per cell line, $p$ value vs Cluster 4 by Unpaired student's t-tests, mean ± standard error of the mean). Source data are available online for this figure.

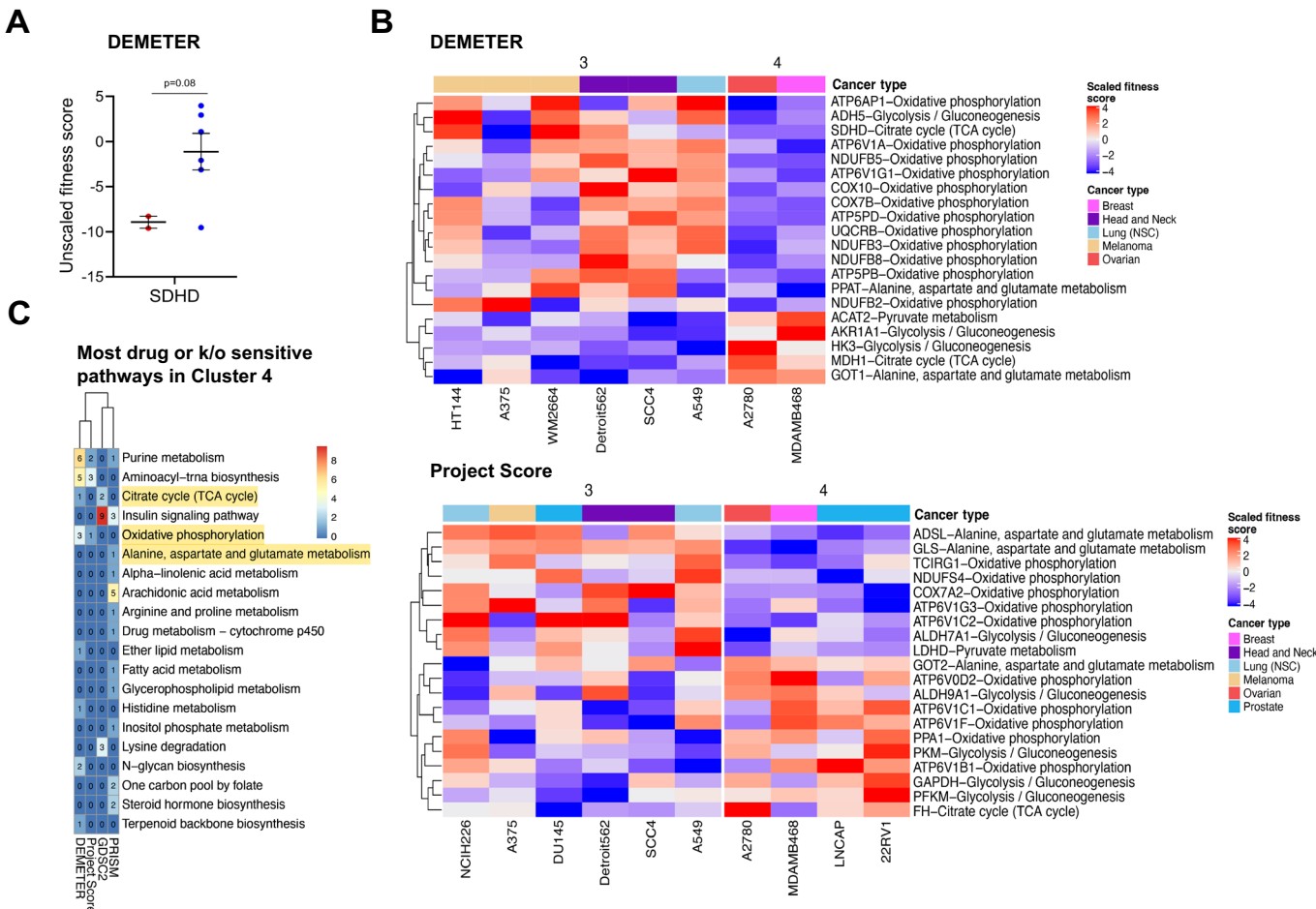

**Figure EV3. Loss of function and drug database screen validation of pathway-centric metabolite ratio clusters.**

(A) Genes related to the glucose utilization phenotype of C4 with trends of greater sensitivity to gene knockouts. (Unpaired student's t-test, ns $p > 0.05$, mean ± standard error of the mean, $n = 2$–6). (B) Top 20 gene knockouts and associated KEGG pathways with the greatest fitness scores between C4 and C3 in DEMETER and Project Score databases from glycolysis, TCA cycle, pyruvate metabolism, glycolysis/gluconeogenesis, and alanine, aspartate, and glutamate metabolism pathways. (C) Metabolic pathways associated with the top 20 drug and gene knockout sensitives for Cluster 4 for each database. Drugs and pathways matching C4 vulnerability model (yellow) are highlighted. Source data are available online for this figure.

## A

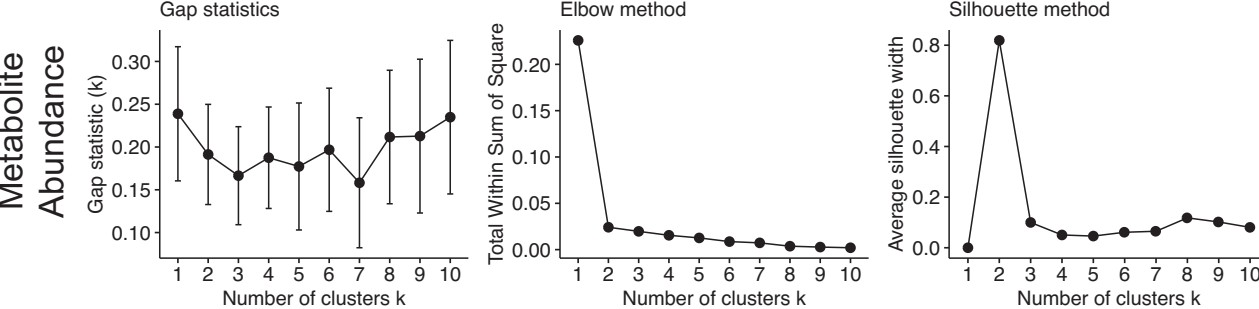

## B

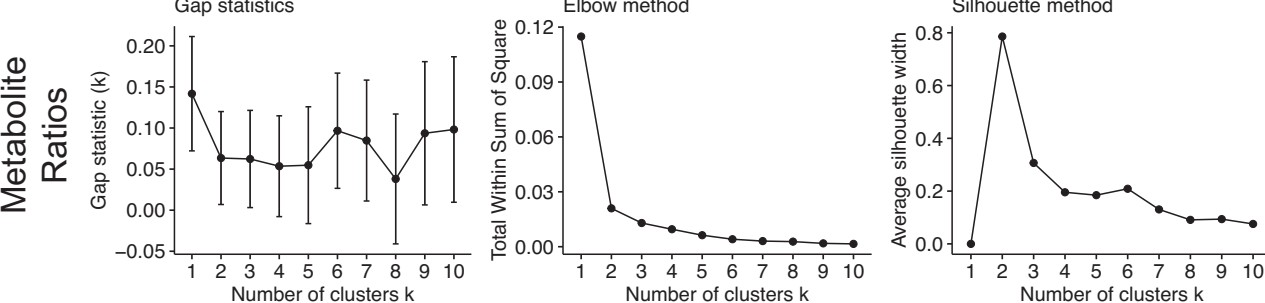

**Figure EV4.  Optimal cluster determination of metabolite and pathway-centric metabolite ratios using published data.**

(**A**) The gap statistic (left), the total within sum of squares using the Elbow method (middle), and the average silhouette width for different number of clusters (right) for metabolite abundance-based clusters. (**B**) The gap statistic (left), the total within sum of squares using the Elbow method (middle), and the average silhouette width for different number of clusters (right) for pathway-centric metabolite ratio-based clusters. Source data are available online for this figure.

