## [Peer Review File · Molecular Systems Biology]

Pathway metabolite ratios reveal distinctive glutamine metabolism in a subset of proliferating cells

Nancy Santiappillai, Yue Cao, Mariam Hakeem-Sanni, Jean Yee Hwa Yang, Lake-Ee Quek, and Andrew Hoy

Corresponding author(s): Andrew Hoy (andrew.hoy@sydney.edu.au)

Review Timeline:

Submission Date:	22nd Feb 24
Editorial Decision:	31st Mar 24
Revision Received:	17th Jun 24
Editorial Decision:	10th Jul 24
Revision Received:	12th Feb 25
Editorial Decision:	11th Mar 25
Revision Received:	21st Mar 25
Accepted:	27th Mar 25

Editor: Poonam Bheda

Transaction Report:

31st Mar 2024

Manuscript Number: MSB-2024-12275

Title: Pathway metabolite ratios reveal distinctive glutamine metabolism in a subset of proliferating cells

Dear Dr Hoy,

Thank you again for submitting your work to Molecular Systems Biology. We have now heard back from the three reviewers who agreed to evaluate your study. As you will see below, the reviewers appreciate that the presented approach addresses a relevant problem. However, they raise a series of concerns, which we would ask you to address in a major revision.

I think that the recommendations of the reviewers are rather clear and I therefore do not see the need to repeat the comments listed below. All issues raised would need to be satisfactorily addressed. Please let me know in case you would like to discuss in further detail any of the reviewer comments, I would be happy to schedule a call.

We require:

4) A .docx formatted letter INCLUDING the reviewers' reports and your detailed point-by-point responses to their comments. As part of the EMBO Press transparent editorial process, the point-by-point response is part of the Peer Review File (PRF), which will be published alongside your paper.

5) A complete author checklist, which you can download from our author guidelines (<https://www.embopress.org/page/journal/17574684/authorguide#submissionofrevisions>). Please insert information in the checklist that is also reflected in the manuscript. The completed author checklist will also be part of the PRF.

6) Please note that all corresponding authors are required to supply an ORCID ID for their name upon submission of a revised manuscript.

7) It is mandatory to include a 'Data Availability' section after the Materials and Methods. Before submitting your revision, primary datasets produced in this study need to be deposited in an appropriate public database, and the accession numbers and database listed under 'Data Availability'. Please remember to provide a reviewer password if the datasets are not yet public (see <https://www.embopress.org/page/journal/17574684/authorguide#dataavailability>).

In case you have no data that requires deposition in a public database, please state so in this section. Note that the Data Availability Section is restricted to new primary data that are part of this study. This study includes no data deposited in external repositories.

8) For data quantification: please specify the name of the statistical test used to generate error bars and P values, the number (n) of independent experiments (specify technical or biological replicates) underlying each data point and the test used to calculate p-values in each figure legend. The figure legends should contain a basic description of n, P and the test applied. Graphs must include a description of the bars and the error bars (s.d., s.e.m.). Please provide exact p values.

9) Our journal encourages inclusion of *data citations in the reference list* to directly cite datasets that were re-used and obtained from public databases. Data citations in the article text are distinct from normal bibliographical citations and should directly link to the database records from which the data can be accessed. In the main text, data citations are formatted as follows: "Data ref: Smith et al, 2001" or "Data ref: NCBI Sequence Read Archive PRJNA342805, 2017". In the Reference list, data citations must be labeled with "[DATASET]". A data reference must provide the database name, accession number/identifiers and a resolvable link to the landing page from which the data can be accessed at the end of the reference.

Further instructions are available at .

<https://www.embopress.org/page/journal/17574684/authorguide#expandedview>

11) For more information: There is space at the end of each article to list relevant web links for further consultation by our readers. Could you identify some relevant ones and provide such information as well? Some examples are patient associations, relevant databases, OMIM/proteins/genes links, author's websites, etc...

12) Author contributions: CRediT has replaced the traditional author contributions section because it offers a systematic machine readable author contributions format that allows for more effective research assessment. Please remove the Authors Contributions from the manuscript and use the free text boxes beneath each contributing author's name in our system to add specific details on the author's contribution. More information is available in our guide to authors.

13) Disclosure statement and competing interests: We updated our journal's competing interests policy in January 2022 and request authors to consider both actual and perceived competing interests. Please review the policy

<https://www.embopress.org/competing-interests> and update your competing interests if necessary.

14) Every published paper now includes a 'Synopsis' to further enhance discoverability. Synopses are displayed on the journal webpage and are freely accessible to all readers. They include a short stand first (maximum of 300 characters, including space) as well as 2-5 one-sentences bullet points that summarizes the paper. Please write the bullet points to summarize the key NEW findings. They should be designed to be complementary to the abstract - i.e. not repeat the same text. We encourage inclusion of key acronyms and quantitative information (maximum of 30 words / bullet point). Please use the passive voice. Please attach these in a separate file or send them by email, we will incorporate them accordingly.

Share synopsis text and image, as well as eTOC:

Please note that these would be the final versions and changes during proofing are usually not allowed

15) As part of the EMBO Publications transparent editorial process initiative (see our Editorial at <http://embomolmed.embopress.org/content/2/9/329>), Molecular Systems Biology Medicine will publish online a Peer Review File (PRF) to accompany accepted manuscripts.

In the event of acceptance, this file will be published in conjunction with your paper and will include the anonymous referee reports, your point-by-point response and all pertinent correspondence relating to the manuscript. Let us know whether you agree with the publication of the PRF and as here, if you want to remove or not any figures from it prior to publication.

Please note that the Authors checklist will be published at the end of the PRF.

Molecular Systems Biology has a "scooping protection" policy, whereby similar findings that are published by others during review or revision are not a criterion for rejection. Should you decide to submit a revised version, I do ask that you get in touch after three months if you have not completed it, to update us on the status.

I look forward to receiving your revised manuscript.

Yours sincerely,

Poonam Bheda, PhD
Scientific Editor
Molecular Systems Biology

Reviewer #1:

The manuscript by Santiappillai et al presents a compelling and new picture of pan cancer metabolism. In contrast to preceding papers, they use our knowledge of metabolic pathways as a prior to enable more insightful cross-comparison. This draws out novel insights and elevates their analysis.

The manuscript is well written and I believe the conclusions sound. The paper would however benefit from the addition of direct cross comparison with published datasets. As the number of papers in the area expand the impact of different technical choices (for example in data processing prior to analysis), and reuse of the data from others would strengthen confidence in the robustness of the conclusions.

Related to this authors must make underlying data from the paper available on publication. At a minimum it should be either deposited or included as supplementary information to this paper. Better would to follow approaches from other papers in the field and present the data through a web dashboard for public access. Similarly code used in data processing needs to be made available.

Finally, statistical tests in the manuscript should be improved. Authors must confirm that where appropriate (e.g. Fig 3C, 4C) that multiple test correction has been performed. If not, the thresholds need to be recalculated and results replotted. Similarly statistical tests should be performed to confirm conclusions drawn on U13-C experiments. Finally, authors should attempt using clustering approaches that enable statistical testing of different cluster separation given the importance to this manuscript (e.g. sigclust).

Reviewer #2:

Santiappillai et al. measure metabolite abundances in 57 cell lines, seeking to identify function-related metabolic signatures and differences among groups of cell lines. The scale of the study is intermediate but reasonable for the conceptual work the authors intend to do. I am not convinced about the novelty of the approach though, as the authors reference similar approaches themselves, and also do not make efforts to present their analysis approach like one would do with a new method (i.e. key performance characteristics and validation are missing, see comments below). Although the authors specifically state that "normal epithelial-derived" cell lines were included, this factor is not further mentioned. The most major concern is that the authors don't seem to have investigated the statistical error in the metabolite ratios, or demonstrate robustness with respect to which pathway precursor is used as a reference.

Specific comments:

** In the last section of the introduction, the authors should explain and justify the "conceptual innovation" they claim to introduce, rather than listing a bunch of results that should anyways be described in detail in the results section. At least the pros and cons of a pathway-centric approach, previous solutions, potential problems with previous approaches and the specific improvement the authors wish to make with their approach should be explained.

** In the first part of the results section, the authors should briefly name the type of measurement (technology, chromatographic separation, if any) that was done to "quantify metabolite levels of high-flux pathways", including how many metabolites were measured and used for analysis. Is the data peak areas/heights or concentrations? It should be mentioned since it affects the meaning of the later metabolite ratios.

** Also, the authors go straight into clustering cell lines, but the underlying rationale and/or meaning of clusters isn't explained. In particular, both here and in the following section where the authors finally address the pathway-centric approach, the authors refer to clusters, but it is poorly differentiated, how the two types/bases for clustering are to be interpreted. Please explain explicitly, after all the paper claims to present an innovative approach, therefore it should be thoroughly introduced and also tested for its validity.

** The authors generally seem to rush a bit through the first section, to get to their preferred method based on ratios. The data is only superficially described, and it is not clear what the authors mean by "strong metabolite interaction" that "are often localized at the reaction level".

** Figure 1C: Why was such a wide range of culture media used? Why did some cultures have enzalutamide added to the medium, i.e. what is the significance thereof? Same for "10% CCS".

** In section 2, where the authors transform metabolite measurements into pathway ratios, I miss a demonstration that the transformation yields informative/expected patterns, i.e. some sort of validation of the approach. In particular, I miss a justification for why the authors think the ratio approach yields "more true" differences between cell lines, as compared to the raw metabolite measurements presented in section 1. This should be presented before diving into the interpretation (lines 105ff).

** Related to the point above, the question arises how sensitive the ratios are to the choice of upstream metabolite? For example, while glucose metabolism is directly linked to pyruvate levels, glutamine more directly feeds into 2-oxoglutarate levels. In the comparative analysis of glucose and glutamine metabolism starting in line 176, did the authors switch between the two types of ratios (i.e. relative to pyruvate vs. relative to 2-oxoglutarate for glutamine), and in either case, would the conclusion change?

** An even more major point is, if the measurement of a pathway's upstream precursor is fraught with high measurement uncertainty, (how) is the error propagated? Pyruvate can be tricky to accurately measure, for example, but much of the discussed TCA cycle-related patterns is based on pyruvate levels.. What is the typical error of the ratio values? How is the

significance of the differences assessed/accounted for in the clustering?

** In line 131 the authors start by claiming they addressed the major limitation in metabolomics, that "we cannot infer function/Flux" - but it's not clear (nor am I convinced) that they have solved the limitation? No validation or verification is presented.

** Line 133: There seems to be some leftover text from manuscript editing "(from where??)".

** In lines 147f the authors state that the "greater lactate to pyruvate ratio in Cluster 4 [...] was unlikely driven by excess NADH relative to NAD+" but later in line 193 they state that "the increase in lactate production is a consequence of cells using pyruvate as a redox sink to regenerate NAD+" ... please check and revise to be clear and avoid contradiction/misunderstandings.

** In lines 195f the authors describe the observations in data from U13C tracing experiments, which "eliminate the role of glucose metabolism in explaining the higher TCA metabolite to pyruvate ratios in Cluster 4 cells". It seems inconsistent with the interpretation of the data in Figure 3, and to refute the model in Figure 3K? If this is true, perhaps the model in Figure 3K should be made less prominent so as to avoid that readers perceive it as confirmed.

** In line 287, it's unclear what the authors mean by "... hypotheses that could be evaluated with functional assessment."?

Reviewer #3:

In this work, Santiappillai et al. apply ratiometric analyses of high-flux metabolic pathways to generate new insights from metabolomics data in cultured cancer cells. Standard metabolomics analysis provides a comprehensive assessment of metabolite abundance within a sample. A key limitation is that these abundances do not necessarily provide insight into mechanistic changes, since metabolomics data alone does not directly correlate with pathway activity. By applying product/precursor ratios the authors generate biologically meaningful insights into pathway activity that are otherwise hidden in standard metabolomics analysis.

To achieve this, the authors performed metabolomics analyses across a broad panel of cancer and epithelial cell lines, representing multiple tissue types. Using metabolite ratios between reactant and product metabolites, the authors identify unique metabolic clusters across different cancer types, independent of culture conditions and oncogenic drivers. These signatures correlate with in silico analyses of loss of function and drug screens. This work represents a key conceptual advance in metabolomics analysis, and offers a potential strategy to infer relevant metabolic activity from these data. Altogether, this work highlights a pathway-driven approach to interrogating metabolic data.

Overall, the authors provide a conceptual advance in the analysis/interpretation of metabolomics data. The data presented are interesting and generally support the authors conclusions. However, there are a few key caveats of this approach that should be discussed further, and there are some data/experiments that would strengthen some of the overall conclusions.

Major Points

1. This work concludes that glutamine contribution to the TCA cycle and OXPHOS distinguishes Clusters 3 and 4. Increasing evidence suggests that non-physiological culture conditions may cause this high glutamine catabolism in some cells (Muir et al, eLife 2017), and that the sensitivity of some cell lines to glutamine inhibition is a product of cell culture that does not translate to in vivo systems (Davidson et al, Cell Metabolism, 2016). The authors should comment on some of the limitations or caveats of extrapolating these in vitro approaches to in vivo, and perhaps speculate on how this strategy might be applied to tumors in vivo.

2. The functional validation experiments are an important aspect of this paper and some further clarifications are needed.

-Figure 3F: The authors show that nearly 100% of glucose and glutamine has been consumed in both C4 and C3 cultures after 24 hours. Two concerns with this data are a) have the cells become metabolically stressed and overconfluent, thus altering the metabolic profiles of these clusters? And b) might proliferation differences across the clusters influence the metabolism at this time point? Calculating glucose/glutamine consumption rates (e.g., glucose consumption/cell #/hour), independent of proliferation differences or nutrient depletion, would be a more compelling comparison between these clusters.

-Figure 3 I/J: In C4, it is difficult to rationalize how [13C]glutamine tracing yields ~5% enrichment of phosphoenolpyruvate, yet its downstream (gluconeogenic) product, FBP is 20% enriched. Can the authors comment on this?

3. Lactate dehydrogenase D (LDHD) sensitivity in Cluster 4 cells (Figure SA3) is not compelling evidence for differences in sensitivity to lactate metabolism. LDHD encodes the D-isomer of LDH, and acts to convert D-lactate to pyruvate. It is the LDHA/B forms that are expressed in most mammalian tissue, and produces L-lactate, the most abundant version of the metabolite in mammals. To make the conclusion that lactate metabolism sensitivity is different between the clusters, can the authors provide evidence that LDHA/B (or, the lactate transporters MCT1-4) are differentially sensitive between the clusters?

4. In the GDSC2 data (Figure 4E) the authors have two drugs AGI-5987 (which targets mutant IDH1, not IDH2) and AG-6780, which inhibits mutant IDH2. Since these drugs were designed for mutant IDH, do any of the examined cell lines harbor IDH

mutations? Or were these drugs used at concentrations to induce off-target effects on WT IDH? In either case, this data does not directly support the conclusion that C4 cells are more vulnerable to drugs targeting the TCA cycle. Are there other drug-sensitivity datasets that more directly support this conclusion? Alternatively, the authors could perform a small, targeted drug panel in a subset of C3 and C4 lines to confirm this conclusion.

Minor Points

-The authors performed a series of metabolomics experiments across multiple cell lines and over multiple years due to Covid. Correcting for batch effect is crucial, and is an important point for the readers. Could the authors provide more details in the methods section on the normalization method used? Were any considerations necessary in applying an RNASeq normalization method to metabolomics data?

-Please review the units used throughout the methods section. Often "mL" is used when it is likely to be " μ L".

-Line 133: Writing/editing remark I believe the authors forgot to remove. (From where??)

We thank the reviewers for generously taking the time to review our manuscript and for their constructive feedback.

In our revised manuscript, we have added large amounts of new text to improve clarity, provide more insights into our rationale and interpretation of our results, additional graphs related to the statistical analyses of our stable isotope tracing data, and details of the methods used as suggested by the reviewers.

A point-to-point responses to the reviewers' carefully considered comments and questions are provided below.

Reviewer #1:

1. The manuscript is well written and I believe the conclusions sound. The paper would however benefit from the addition of direct cross comparison with published datasets. As the number of papers in the area expand the impact of different technical choices (for example in data processing prior to analysis), and reuse of the data from others would strengthen confidence in the robustness of the conclusions.

Response: we thank the reviewer for their kind comments in this point and overall summary. Upon reflection, we are disappointed that we did not think of validating our observations reported in Figure 2 and the application of pathway-centric ratios in existing data, similar to our *in silico* approaches using existing loss-of-function dataset.

We have closely examined the available data for those studies we cite in our introduction and others (Benedetti et al, 2023; Cherkaoui et al, 2022; Jain et al, 2012; Li et al, 2019; Mullen & Singh, 2023; Ortmayr et al, 2019; Shorthouse et al, 2022) and determined that they are not in a format that allows us to generate ratios or compare our results. Specifically, many datasets are already processed/transformed (log transformed, relative abundances) or CORE (consumption/release) profiling and/or have limited overlap with our collection of cell lines. Metabolite intensities/ratios from different sources are difficult to compare directly and can lead to wrong results, unless there is suitable model for standardisation, i.e., to compensate for technical batch effects. For our case of calculating ratios in particular, sample variance and measurement errors amplify noise. This means the precision tend to be poorer for metabolite ratios that are lower and that are derived from lower signal intensities. Thus, it can be important to compare published data that reports molar concentrations and not relative intensities.

2. Related to this authors must make underlying data from the paper available on publication. At a minimum it should be either deposited or included as supplementary information to this paper. Better would to follow approaches from other papers in the field and present the data through a web dashboard for public access. Similarly code used in data processing needs to be made available.

Response: we apologise for this sub-optimal experience for the reviewer. In our recent experiences, every journal/publisher has their own preferences for handling raw/underlying data, and it is tiresome to have to move from one platform to another; especially with the high possibility of an editorial desk rejection. After receiving the Editor's and Reviewers' comments, we received a request from the journal regarding the underlying data. We have included the required information on how to access the data and codes in our revised manuscript.

3. Finally, statistical tests in the manuscript should be improved. Authors must confirm that where appropriate (e.g. Fig 3C, 4C) that multiple test correction has been performed. If not, the thresholds need to be recalculated and results replotted. Similarly statistical tests should be performed to confirm conclusions drawn on U13-C experiments. Finally, authors should attempt using clustering approaches that enable statistical testing of different cluster separation given the importance to this manuscript (e.g. sigclust).

Response: We did not perform multiple test corrections for Figs 3C and 4C because our end point is to show these individual features are different between C3 and C4, and the direction of change are consistent among these features. We have removed “multiple” from the manuscript when stating which statistical test was used to assess significance for these data (e.g. lines 147, 172-3, 214-215).

We have performed additional analyses on the U13-C glutamine experiments reported in Figures 3I and J. We have included those analyses as Figures EV2F & G.

The suggested method sigclust is design to test whether one should “further split” a node inside a hierarchical clustering context and here, the problem is better phrase as “estimating the optimal number of clusters”. To address this used standard approaches for determining the optimal number of clusters such as WSS and Silhouette Distance as implemented in the R package “factoextra”. These approaches involved selecting the optimal “k” using the elbow method with the within sum of square (WSS) and the Silhouette Distance. The WSS measures the compactness of the clustering, meaning lower is better. Silhouette measures the consistency of the data within a cluster, meaning higher is better. Examine these two statistics jointly (Figures for Reviewer 1), we determined that when $k = 5$, this is a knee point when the WSS starts to plateau, and it also has a relatively high silhouette width.

Figure for Reviewer 1: Line plots showing the total within sum of squares (top panel) and average silhouette width (bottom panel) for different number of clusters (x-axis).

Reviewer #2:

Santiappillai et al. measure metabolite abundances in 57 cell lines, seeking to identify function-related metabolic signatures and differences among groups of cell lines. The scale of the study is intermediate but reasonable for the conceptual work the authors intend to do. I am not convinced about the novelty of the approach though, as the authors reference similar approaches themselves, and also do not make efforts to present their analysis approach like one would do with a new method (i.e. key performance characteristics and validation are missing, see comments below). Although the authors specifically state that "normal epithelial-derived" cell lines were included, this factor is not further mentioned.

The most major concern is that the authors don't seem to have investigated the statistical error in the metabolite ratios, or demonstrate robustness with respect to which pathway precursor is used as a reference.

Response: we thank the reviewer for their comments and overall summary. We have attempted to address the major concerns (statistical error, pathway precursor selection, others) below. Here, we wanted to address the comment related to a new method and related performance measures. We hope that our manuscript does not give the impression that we are presenting a new method, certainly not in the classical sense. Ultimately, we believe that our story centres on transforming data (like fold-change, % change, IC50 etc) to gain new, targeted physiological insights and showcases the benefits through complementary methods (e.g. stable-isotope tracing, *in silico* analyses). We have taken onboard this possible interpretation of our manuscript and attempted to refine our messaging.

Specific comments:

1. In the last section of the introduction, the authors should explain and justify the "conceptual innovation" they claim to introduce, rather than listing a bunch of results that should anyways be described in detail in the results section. At least the pros and cons of a pathway-centric approach, previous solutions, potential problems with previous approaches and the specific improvement the authors wish to make with their approach should be explained.

Response: we appreciate the reviewer's view on what information should be in the final paragraph of the introduction. We were heavily inspired by similar cancer cell line metabolomic publications and recent articles in *Molecular Systems Biology*. In our revised manuscript, we have altered the latter parts of the Introduction (Lines 45-47 and 51-61) to explain and justify our belief that our approach is a conceptual innovation. In particular, we have provided more detail on the approach used by Cherkaoui and colleagues (2022) to assess pathway activity and compared this to our approach.

2. In the first part of the results section, the authors should briefly name the type of measurement (technology, chromatographic separation, if any) that was done to "quantify

metabolite levels of high-flux pathways", including how many metabolites were measured and used for analysis. Is the data peak areas/heights or concentrations? It should be mentioned since it affects the meaning of the later metabolite ratios.

Response: we prioritised presenting detailed descriptions of methodological information predominantly in the Materials and Methods and included some key details in Figure 1A. However, we can see the benefit for additional details being provided in the Results section. We have included a brief description of the method used (LC-MS/MS), the coverage (50 metabolites) and analyses used on lines 76-77.

3. Also, the authors go straight into clustering cell lines, but the underlying rationale and/or meaning of clusters isn't explained. In particular, both here and in the following section where the authors finally address the pathway-centric approach, the authors refer to clusters, but it is poorly differentiated, how the two types/bases for clustering are to be interpreted. Please explain explicitly, after all the paper claims to present an innovative approach, therefore it should be thoroughly introduced and also tested for its validity.

Response: we value the very fair criticisms of these aspects of our manuscript. In our revised manuscript, we have expanded purpose for clustering cells, our description of these results and our interpretation of the two sets of clusters (lines 81-91 and 103-105).

4. The authors generally seem to rush a bit through the first section, to get to their preferred method based on ratios. The data is only superficially described, and it is not clear what the authors mean by "strong metabolite interaction" that "are often localized at the reaction level".

Response: Similar to point 3, this is a valid critique of this section of the manuscript. We now describe the results presented in Figure 1 and associated data in more detail. Also, we have attempted to be clearer in the major limitation of the outcomes reported in Figure 1 to assist in the crafting of the rationale for transforming our metabolite concentration data into pathway-based ratios in our revised manuscript (lines 89-101).

5. Figure 1C: Why was such a wide range of culture media used? Why did some cultures have enzalutamide added to the medium, i.e. what is the significance thereof? Same for "10% CCS".

Response: On reflection, this is a question that should have been able to be answered in our original submission. We have included additional statements that explain our selection of cell culture conditions in the Material and Methods section (lines 333-339).

6. In section 2, where the authors transform metabolite measurements into pathway ratios, I miss a demonstration that the transformation yields informative/expected patterns, i.e. some sort of validation of the approach. In particular, I miss a justification for why the authors think the ratio approach yields "more true" differences between cell lines, as compared to the raw

metabolite measurements presented in section 1. This should be presented before diving into the interpretation (lines 105ff).

Response: I am not certain that we conveyed a message that the ratio approach yields "more true" differences between cell lines. Our belief is more that the clustering by metabolite concentrations fails to provide pathway specific information (as evident by the distribution of upper glycolysis or pentose phosphate pathway metabolites in Fig 1) and that by arranging these data into pathway silos facilitates the identification of differences in pathways. We are not advocating that there is a critical issue with pipeline analyses of metabolite abundance/concentration, especially for research questions centred on changes in metabolites. It's just that in our hands, it is hard to gain insights into pathway behaviour of cells if, for example, FBP is at the top and glucose is at the bottom of the hierarchical clustering heatmap.

7. Related to the point above, the question arises how sensitive the ratios are to the choice of upstream metabolite? For example, while glucose metabolism is directly linked to pyruvate levels, glutamine more directly feeds into 2-oxoglutarate levels. In the comparative analysis of glucose and glutamine metabolism starting in line 176, did the authors switch between the two types of ratios (i.e. relative to pyruvate vs. relative to 2-oxoglutarate for glutamine), and in either case, would the conclusion change?

Response: The selection of upstream metabolite is very important, and we do acknowledge this point in our limitation paragraph of the Discussion (lines 293 onwards in revision). Our approach was to assess metabolite concentrations in pathways, where all quantified members of the pathway were included, not switching denominators, or exclude reactions, and their corresponding substrate or product, which were present in multiple metabolic pathways as was performed by Cherkaoui *et al.* (2022).

In Figure 2 and S1, we report the ratios of TCA cycle and other related metabolites using glutamine as the denominator but did not deeply explore this data. Figure for Reviewer 2 shows that there are some differences in the TCA cycle to glutamine ratios, but not the 2-oxoglutarate-glutamine ratio. Importantly, the direct of change in this data is the same as what is reported in Figure 3C (pyruvate ratios). Whilst the magnitude of difference between Cluster 3 and 4 are generally smaller when using glutamine as the denominator compared to pyruvate, we believe this does not alter our conclusions.

Figure for Reviewer 1: Abundance ratios of TCA cycle metabolites to glutamine (C4 n=10, C3 n=19, multiple unpaired t-tests, *p<0.05, error min to max values).

8. An even more major point is, if the measurement of a pathway's upstream precursor is fraught with high measurement uncertainty, (how) is the error propagated? Pyruvate can be tricky to accurately measure, for example, but much of the discussed TCA cycle-related patterns is based on pyruvate levels. What is the typical error of the ratio values? How is the significance of the differences assessed/accounted for in the clustering?

Response: We believe that this is very fair criticism, but one that applies to all 'omics data analyses that use conventional dimensional reduction, clustering, and over-presentation methodologies, especially genomics/transcriptomics.

In response to this concern, as pyruvate is the key denominator used in our study, we evaluated %RSD of raw pyruvate measurements, and found it to be 11% for the QC (pool, 10 injections) samples, 74% for all samples (78 samples), and an average of 14% for the technical replicates (26 cell lines, n=3 each). This means the error associated with pyruvate measurements is generally small relative to observed differences between cell lines. For succinate (one of the numerators), this was 8%, 62% and 10% for QC, all samples and technical replicates, respectively. Thus measurement errors were relatively minor and were not included in the clustering analyses, which is typical of these approaches.

As for statistical significance in our clusters, this was also raised by Reviewer 1 (point 3). Briefly, we did not perform statistical tests (e.g. sigclust or similar), but rather used standard approaches for determining the optimal number of clusters.

9. In line 131 the authors start by claiming they addressed the major limitation in metabolomics, that "we cannot infer function/Flux" - but it's not clear (nor am I convinced) that they have solved the limitation? No validation or verification is presented.

Response: this is a very fair criticism of our original submission. We have re-written this sentence (now lines 138-142 in revised submission) to improve clarity and better align with the analyses we performed.

10. Line 133: There seems to be some leftover text from manuscript editing "(from where??)".

Response: This has been addressed.

11. In lines 147f the authors state that the "greater lactate to pyruvate ratio in Cluster 4 [...] was unlikely driven by excess NADH relative to NAD+" but later in line 193 they state that "the increase in lactate production is a consequence of cells using pyruvate as a redox sink to regenerate NAD+" ... please check and revise to be clear and avoid contradiction/misunderstandings.

Response: thank you for identifying this contradiction. We have altered the latter sentence (now lines 175-176 and of revision) to avoid any misunderstandings.

12. In lines 195f the authors describe the observations in data from U13C tracing experiments, which "eliminate the role of glucose metabolism in explaining the higher TCA metabolite to pyruvate ratios in Cluster 4 cells". It seems inconsistent with the interpretation of the data in Figure 3, and to refute the model in Figure 3K? If this is true, perhaps the model in Figure 3K should be made less prominent so as to avoid that readers perceive it as confirmed.

Response: upon reflection, we did not choose our words as wisely as we could have, and so have recrafted this sentence and related text to improve clarity and accuracy of our description of these results (lines 183-185).

13. In line 287, it's unclear what the authors mean by "... hypotheses that could be evaluated with functional assessment."?

Response: we have added additional information in an attempt to improve the precision of this text (lines 250-252).

Reviewer #3:

1. This work concludes that glutamine contribution to the TCA cycle and OXPHOS distinguishes Clusters 3 and 4. Increasing evidence suggests that non-physiological culture conditions may cause this high glutamine catabolism in some cells (Muir et al, eLife 2017), and that the sensitivity of some cell lines to glutamine inhibition is a product of cell culture that does not translate to *in vivo* systems (Davidson et al, Cell Metabolism, 2016). The authors should comment on some of the limitations or caveats of extrapolating these *in vitro* approaches to *in vivo*, and perhaps speculate on how this strategy might be applied to tumors *in vivo*.

Response: we appreciate this view and acknowledge the limitations of traditional cell culture media composition. The strength of our approach was that we accepted media composition as a source of diversity. In both Figs 1 & 2 we show that the clustering of cells based upon metabolism was not simply explained by media composition.

As suggested by the reviewer, we have added text related to the limitations of extrapolating *in vitro* approaches to *in vivo* given the reports from Muir et al and Davidson et al and cell culture media conditions, and cautiously speculated on how our metabolite-ratio approach could be applied to metabolomic data from tumour tissue (lines 305-317).

2. The functional validation experiments are an important aspect of this paper and some further clarifications are needed.

-Figure 3F: The authors show that nearly 100% of glucose and glutamine has been consumed in both C4 and C3 cultures after 24 hours. Two concerns with this data are a) have the cells become metabolically stressed and overconfluent, thus altering the metabolic profiles of these clusters? And b) might proliferation differences across the clusters influence

the metabolism at this time point? Calculating glucose/glutamine consumption rates (e.g., glucose consumption/cell #/hour), independent of proliferation differences or nutrient depletion, would be a more compelling comparison between these clusters.

Response: these are very valid concerns and awareness of rapid consumption of media nutrients in these kinds of experiments is one we are attempting to make others in the field more aware of. Here, I hope we can sufficiently address the above concerns.

- A) The experimental conditions used to assess glucose and glutamine consumption and lactate production were different to the conditions we used to perform our targeted metabolic and stable isotope tracing experiments. Specifically, the consumption/production experiments were performed in the same media composition as we used for the stable-isotope tracing experiments (5 mM ¹²C-glucose, 1 mM ¹²C-glutamine, 150 μM ¹²C-palmitate). Whereas our targeted metabolomic measures (e.g. reported in Figs 1 &2) were in cells cultured in their growth media at a final confluency of ~70-80% - cell health and confluency were checked prior to sample processing.
- B) Cells were cultured in stable isotope containing media for only 6 hours, and so we believe that any differences observed are independent of proliferation rates. Notably, the glucose and glutamine levels in the ¹³C-supplemented media 6 hours were for nearly all cells >50% of starting (Fig S2B).
- C) Whilst we do not believe that our data can be explained by differences in proliferation rates, we can add this information if the reviewer believes that this is important.

-Figure 3 I/J: In C4, It is difficult to rationalize how [¹³C]glutamine tracing yields ~5% enrichment of phosphoenolpyruvate, yet its downstream (gluconeogenic) product, FBP is 20% enriched. Can the authors comment on this?

Response:

Indeed this disparity had previously raised our concern, but we have double checked the underlying data and could not find data artefact contributing to this observation. A plausible mechanism for ¹³C enrichment of FBP (conversion from glutamine) to have seemingly bypassed PEP could be explain by combination of PEPCK localisation, which could be cytoplasmic or mitochondrial, and metabolic channelling (metabolons). It is possible that a small pool of glutamine-derived mitochondria PEP produced by mtPEPCK is channelled into the glucogenic pathway, with the intermediates not well-mixed with the rest of glycolytic metabolites. While this observation itself is interesting, this was not a significant finding due to a large variance in m+3 FBP (Figure 3J). Nonetheless, together with lactate and PEP enrichment, and C4 TCA intermediates generally being more abundant relative to pyruvate, these results from Figure 3 together reinforced that there was a greater glutamine cataplerosis in C4 cells, with glutamine carbon converted into pyruvate.

3. Lactate dehydrogenase D (LDHD) sensitivity in Cluster 4 cells (Figure SA3) is not compelling evidence for differences in sensitivity to lactate metabolism. LDHD encodes the D-isomer of LDH, and acts to convert D-lactate to pyruvate. It is the LDHA/B forms that are expressed in most mammalian tissue, and produces L-lactate, the most abundant version of the metabolite in mammals. To make the conclusion that lactate metabolism sensitivity is different between the clusters, can the authors provide evidence that LDHA/B (or, the lactate transporters MCT1-4) are differentially sensitive between the clusters?

Response: this is a very thoughtful critique of our data, and appreciate the alternative explanation. We have examined the LDHA/B in the datasets, and they were not significantly different. Interestingly, we discovered that MCT1 & 4 (*SLC16A1*, *SLC16A4*) are not mapped to glycolysis/gluconeogenesis but efferocytosis in KEGG. As such, these targets fell out of scope early in our curation (see Fig 4A for process). We have gone back into the datasets to examine MCT1 & 4 (*SLC16A1*, *SLC16A4*) in cells that are members of Clusters 3 or 4. There was no difference in sensitivity and so we have removed related text in our revised manuscript.

4. In the GDSC2 data (Figure 4E) the authors have two drugs AGI-5987 (which targets mutant IDH1, not IDH2) and AG-6780, which inhibits mutant IDH2. Since these drugs were designed for mutant IDH, do any of the examined cell lines harbor IDH mutations?

Or were these drugs used at concentrations to induce off-target effects on WT IDH?

In either case, this data does not directly support the conclusion that C4 cells are more vulnerable to drugs targeting the TCA cycle. Are there other drug-sensitivity datasets that more directly support this conclusion?

Alternatively, the authors could perform a small, targeted drug panel in a subset of C3 and C4 lines to confirm this conclusion.

Response: we are grateful for these expert insights by the reviewer. We report mutant status of common oncogenes/tumour suppressors in Figures 1C and S1B. Only a small number of our cells are reported to harbour IHD mutations according to Cellosaurus database, COSMIC database, and Depmap portal, with only 1 cell line in Cluster 4 and no cell lines in Cluster 3. We cannot speak to the design or outcomes other researchers experiments, especially drug concentrations and potential off-target effects. This is, in part, why we chose to test our hypothesis in multiple published datasets that used different approaches/technologies and believe that the power of our results is that they arise from multiple, unbiased loss-of-function screens.

As for other drug-sensitivity datasets, we are aware of Pemovska et al (Nat Commun. 2021 Dec 14;12(1):7190. PMID: 34907165) which was limited to myeloid leukemia cell lines, and Bashi et al (Cancer Discov 2024 May 1;14(5):846-865. PMID: 38456804) which only assessed 47 established anti-cancer drugs that target genome integrity, apoptosis, and the cell cycle. These datasets do not provide an opportunity to test our hypothesis.

As for performing a small, targeted drug experiment, we are not a pharmacology lab but know that this is no small effort to do properly. We are heavily influenced by (Yao *et al*, 2018) who highlighted the differences between the minimal concentration of a drug to maximally inhibit its target reaction and the concentration that kills cells. As such, any drug screen should first screen for target function before cell viability. In any event, we used the loss-of-function screens to test our hypothesis that Clusters 3 and 4 have different sensitivities based on our results, not to identify a potential therapeutic strategy for the member cancer subtypes in these clusters.

Minor Points

-The authors performed a series of metabolomics experiments across multiple cell lines and over multiple years due to Covid. Correcting for batch effect is crucial, and is an important point for the readers. Could the authors provide more details in the methods section on the normalization method used? Were any considerations necessary in applying an RNASeq normalization method to metabolomics data?

Response: we have expanded the description of the methods used for batch correction in the Materials and Methods section of our revised manuscript (lines 377-387). Briefly, we used RUV-III for normalisation (Molania *et al*, 2019). This remove unwanted variation (RUV) approach has been used in multiple omics including metabolite data (De Livera *et al*, 2015). We used the RUV-III version as it can account for replicates in the estimation of unwanted noise and replicate samples are a key part of our experimental design,

To assess the quality of the normalization approach, we examined the variation between cell lines replicates across multiple runs. We achieve this by examining K-means clustering results with Pearson correlation to ensure the same cell lines across multiple runs were clustered in the same group after normalisation.

-Please review the units used throughout the methods section. Often "mL" is used when it is likely to be "µL".

Response: This has been addressed.

-Line 133: Writing/editing remark I believe the authors forgot to remove. (From where??)

Response: This has been addressed.

References for Reviewers

Benedetti E, Liu EM, Tang C, Kuo F, Buyukozkan M, Park T, Park J, Correa F, Hakimi AA, Intlekofer AM *et al* (2023) A multimodal atlas of tumour metabolism reveals the architecture of gene–metabolite covariation. *Nat Metab* 5: 1029-1044

Cherkaoui S, Durot S, Bradley J, Critchlow S, Dubuis S, Masiero MM, Wegmann R, Snijder B, Othman A, Bendtsen C *et al* (2022) A functional analysis of 180 cancer cell lines reveals conserved intrinsic metabolic programs. *Mol Syst Biol* 18

De Livera AM, Sysi-Aho M, Jacob L, Gagnon-Bartsch JA, Castillo S, Simpson JA, Speed TP (2015) Statistical methods for handling unwanted variation in metabolomics data. *Anal Chem* 87: 3606-3615

Jain M, Nilsson R, Sharma S, Madhusudhan N, Kitami T, Souza AL, Kafri R, Kirschner MW, Clish CB, Mootha VK (2012) Metabolite profiling identifies a key role for glycine in rapid cancer cell proliferation. *Science* 336: 1040-1044

Li H, Ning S, Ghandi M, Kryukov GV, Gopal S, Deik A, Souza A, Pierce K, Keskula P, Hernandez D *et al* (2019) The landscape of cancer cell line metabolism. *Nat Med* 25: 850-860

Mullen NJ, Singh PK (2023) Nucleotide metabolism: a pan-cancer metabolic dependency. *Nat Rev Cancer* 23: 275-294

Ortmayr K, Dubuis S, Zampieri M (2019) Metabolic profiling of cancer cells reveals genome-wide crosstalk between transcriptional regulators and metabolism. *Nat Commun* 10

Shorthouse D, Bradley J, Critchlow SE, Bendtsen C, Hall BA (2022) Heterogeneity of the cancer cell line metabolic landscape. *Mol Syst Biol* 18

Yao CH, Liu GY, Wang R, Moon SH, Gross RW, Patti GJ (2018) Identifying off-target effects of etomoxir reveals that carnitine palmitoyltransferase I is essential for cancer cell proliferation independent of beta-oxidation. *PLoS Biol* 16: e2003782

10th Jul 2024

Manuscript Number: MSB-2024-12275R

Title: Pathway metabolite ratios reveal distinctive glutamine metabolism in a subset of proliferating cells

Dear Prof. Hoy,

Thank you again for submitting your revised work to Molecular Systems Biology. We have now heard back from the original three reviewers who evaluated your study. As you will see below, while two of the reviewers are satisfied with the revised manuscript, Reviewer 1 finds comments that the revisions did not address fundamental issues that were previously raised with a few additions in response to your point-by-point response. In a cross-commenting session between the reviewers and the editor, all of the reviewers agreed that multiple test correction should be done and that a justification for number of clusters should be provided, in line with Reviewer 1's comments. The other suggestion by Reviewer 1 to compare to previously published datasets should also be addressed, given that the raw data are indeed available. We would also encourage you to go back through Reviewer 1's previous comments and ensure that you have addressed each concern fully, as Reviewer 1 commented that the revised manuscript did not address their original comments. I must note that Reviewer 2 also agreed with this and commented that the revisions minimally addressed the concerns. We would therefore ask you to address their concerns fully in a revision.

We remind you that we have the following formatting requirements:

1) A .docx formatted version of the manuscript text (including legends for main figures, EV figures and tables). Please make sure that the changes are highlighted to be clearly visible. Alternatively you may choose to submit your manuscript as a LaTeX file.

4) A .docx formatted letter INCLUDING the reviewers' reports and your detailed point-by-point responses to their comments. As part of the EMBO Press transparent editorial process, the point-by-point response is part of the Peer Review File (PRF), which will be published alongside your paper.

5) A complete author checklist, which you can download from our author guidelines (<https://www.embopress.org/page/journal/17574684/authorguide#submissionofrevisions>). Please insert information in the checklist that is also reflected in the manuscript. The completed author checklist will also be part of the PRF.

6) Please note that all corresponding authors are required to supply an ORCID ID for their name upon submission of a revised manuscript.

7) It is mandatory to include a 'Data Availability' section after the Materials and Methods. Before submitting your revision, primary datasets produced in this study need to be deposited in an appropriate public database, and the accession numbers and database listed under 'Data Availability'. Please remember to provide a reviewer password if the datasets are not yet public (see <https://www.embopress.org/page/journal/17574684/authorguide#dataavailability>).

This study includes no data deposited in external repositories.

8) All Materials and Methods need to be described in the main text using our 'Structured Methods' format, which is required for all research articles. According to this format, the Methods section includes a Reagents and Tools Table (listing key reagents, experimental models, software and relevant equipment and including their sources and relevant identifiers) followed by a Methods and Protocols section describing the methods using a step-by-step protocol format. The aim is to facilitate adoption of the methodologies across labs. More information on how to adhere to this format as well as a downloadable template (.docx) for the Reagents and Tools Table can be found in our author guidelines:

<https://www.embopress.org/page/journal/17444292/authorguide#structuredmethods>

9) For data quantification: please specify the name of the statistical test used to generate error bars and P values, the number (n) of independent experiments (specify technical or biological replicates) underlying each data point and the test used to calculate p-values in each figure legend. The figure legends should contain a basic description of n, P and the test applied. Graphs must include a description of the bars and the error bars (s.d., s.e.m.). Please provide exact p values.

10) Our journal encourages inclusion of *data citations in the reference list* to directly cite datasets that were re-used and obtained from public databases. Data citations in the article text are distinct from normal bibliographical citations and should directly link to the database records from which the data can be accessed. In the main text, data citations are formatted as follows: "Data ref: Smith et al, 2001" or "Data ref: NCBI Sequence Read Archive PRJNA342805, 2017". In the Reference list, data citations must be labeled with "[DATASET]". A data reference must provide the database name, accession number/identifiers and a resolvable link to the landing page from which the data can be accessed at the end of the reference. Further instructions are available at .

11) We replaced Supplementary Information with Expanded View (EV) Figures and Tables that are collapsible/expandable online. A maximum of 5 EV Figures can be typeset. EV Figures should be cited as 'Figure EV1, Figure EV2' etc... in the text and their respective legends should be included in the main text after the legends of regular figures.

<https://www.embopress.org/page/journal/17574684/authorguide#expandedview>

13) Author contributions: CRediT has replaced the traditional author contributions section because it offers a systematic machine readable author contributions format that allows for more effective research assessment. Please remove the Authors Contributions from the manuscript and use the free text boxes beneath each contributing author's name in our system to add specific details on the author's contribution. More information is available in our guide to authors.

Share synopsis text and image, as well as eTOC:

Please note that these would be the final versions and changes during proofing are usually not allowed

16) As part of the EMBO Publications transparent editorial process initiative (see our policy here:

https://www.embopress.org/transparent-process#Review_Process), Molecular Systems Biology will publish online a Peer Review File (PRF) to accompany accepted manuscripts.

In the event of acceptance, this file will be published in conjunction with your paper and will include the anonymous referee reports, your point-by-point response and all pertinent correspondence relating to the manuscript. Let us know whether you agree with the publication of the PRF and as here, if you want to remove or not any figures from it prior to publication.

Please note that the Authors checklist will be published at the end of the PRF.

Molecular Systems Biology has a "scooping protection" policy, whereby similar findings that are published by others during review or revision are not a criterion for rejection. Should you decide to submit a revised version, I do ask that you get in touch after three months if you have not completed it, to update us on the status.

I look forward to receiving your revised manuscript.

Yours sincerely,

Poonam Bheda

Poonam Bheda, PhD
Scientific Editor
Molecular Systems Biology

Reviewer #1:

The authors revised manuscript and rebuttal does not address the fundamental issues with the previous draft and they should revise in light of the original comments and the additions below.

Notably with respect to comparison with previous datasets, the works by Cherkaoui and Shorthouse are based on the same dataset (normalised differently), and Cherkaoui shares the raw data on the Massive database (<https://massive.ucsd.edu/ProteoSAFe/dataset.jsp?accession=MSV000087155>), with annotations available (<https://doi.org/10.3929/ethz-b-000511784>). The authors should use this data or clearly state why this unprocessed data is unsuitable.

On statistical tests, the purpose of multiple test correction is to avoid false positives that can arise from performing large numbers of tests, and is not solely required when doing comparisons. This is important here as some of the results appear borderline significant and adjusting the threshold will reduce the risk of this type of error.

Finally, the analysis of clustering remains inadequate, and the presented data does not clearly enable the authors to make the argument that five clusters is a more meaningful split than four or seven. Whilst the authors do not need to use sigclust, they need to better statistically justify the clusters selected as they are fundamental to the paper.

Reviewer #2:

The authors have considered all points raised and have carefully revised the manuscript satisfactorily. Just a final comment that is not intended to hinder publication of this work - the authors have argued well why they think measurement error likely has a minor impact on metabolite ratios. However, I would have liked finding such a comment also in the manuscript, to make the reader aware that it could play a role if one were to adapt the approach to other data.

Reviewer #3:

The manuscript is clear and well-written, and the additional details provide important insight for the reader. The authors clearly demonstrate the utility and conceptual innovation of how metabolomic data transformations can gain new insights into cancer cell biology.

I want to thank the authors for their efforts in addressing these critiques and for their thoughtful and thorough responses. I have no further comments.

We thank the reviewers for generously taking the time again to review our revised manuscript and for their constructive feedback.

In our second revised manuscript, we have added new text to improve clarity related to the methods used to determine the optimal number of clusters and discussed measurement error. We have also re-analysed relevant data using multiple unpaired t-tests corrected for FDR, and updated figures and related text, as suggested by the reviewers. Finally, we analysed the data from Shorthouse *et al.* (2022) and describe the outcomes below.

Point-to-point responses to the reviewers' comments are provided below.

Reviewer #1:

1. The authors revised manuscript and rebuttal does not address the fundamental issues with the previous draft and they should revise in light of the original comments and the additions below.

Notably with respect to comparison with previous datasets, the works by Cherkaoui and Shorthouse are based on the same dataset (normalised differently), and Cherkaoui shares the raw data on the Massive database (<https://massive.ucsd.edu/ProteoSAFe/dataset.jsp?accession=MSV000087155>), with annotations available (<https://doi.org/10.3929/ethz-b-000511784>). The authors should use this data or clearly state why this unprocessed data is unsuitable.

Response: We appreciate the reviewer's views and apologise for not meeting their expectations in our revised manuscript. We believed that we directly and appropriately responded to these original comments but appreciate the reviewer's rationale for these analyses of published data should be attempted. We have now completed Reviewer's recommendation to carry out a direct cross comparison with published data, that is reapplying our data processing workflow on Shorthouse dataset, with the aim of validating our major observations and strengthening confidence in the robustness of the conclusions. Specifically, we performed the following steps:

1. We selected Shorthouse *et al.* (2022), which used primary data published in Cherkaoui *et al.* (2022), as they expanded the metabolite coverage and introduced additional normalisation and filtering. Between Shorthouse (179 cell lines, 1099 metabolites) and ours (57 cell lines, 50 metabolites), 39 metabolites and only 15 cell lines overlap. Upfront, it is critical to highlight that the two metabolomics datasets are strikingly different: **1)** Cherkaoui is untargeted data, reported as ion intensities, which was generated by flow injection (no LC separation, i.e., isomers like lysine and glutamine are not distinguished), whereas our targeted data, reported as concentrations from the included standards for all 50 metabolites, was generated by LC-MS/MS. **2)** in Cherkaoui, ALL cell lines were adapted to RPMI + 10%FCS, whereas we used normal/recommended media for each cell line. **3)** different metabolite extraction protocol and recovery efficiency: Cherkaoui used monophasic 2:2:1 acetonitrile:methanol:water, whereas we used biphasic 1:1:2 methanol:water:chloroform.
2. Next, we log₁₀ transformed and performed quantile normalisation on the Shorthouse data. The first red flag we observed from these analyses was that 16 out of 39 metabolites, common among both datasets, had negative correlation coefficients (Figure for Reviewer 1). We would expect more positive correlations if metabolite profiles are cell line specific. As extrinsic factors have nontrivial impacts on the metabolite profile (Golikov *et al.*, 2022), and knowing our cell culture conditions and

extraction protocol differed from Cherkaoui, this first result raises significant challenges in reconciling both datasets.

Figure for Reviewer 1: Pearson correlation of metabolite concentrations in our dataset compared to wmetabolite ion intensities reported in Shorthouse et al. * indicates decoupled metabolite isomers.

3. Despite this significant discrepancy, we continued and performed hierarchical clustering on Shorthouse et al. abundances of the 39 common metabolites. The combination of K-means clustering, gap statistics, elbow method and silhouette method determined that the optimal number of clusters of cells was 5 (Figure for Reviewer 2).
 - a. We identified that amino acids were more abundant in cells in Clusters 4 and 5 of Shorthouse et al. than in the other Clusters. We did not observe this pattern in our data, which could arise from the use of a common culture media in Shorthouse.
 - b. Cell lines did not cluster according to cancer types, which we report in our manuscript.
 - c. The clusters consisted of different combinations of cell lines (Figure for Reviewer 2) compared to our analyses (revised manuscript Figure 1).
 - d. There was no apparent organization of metabolites into pathways, which is consistent with our metabolite concentration data results.

- e. Overall, both datasets clearly show that metabolites do not cluster by pathway using this approach, supporting a major observation presented in our revised manuscript.

Figure for Reviewer 2: Heatmap of scaled metabolite expression across central carbon and amino acid metabolism within the common cell lines to Santiappillai et al. and Shorthouse et al. (n=15). Cell lines colour-coded by cancer type and tissue type.

4. We proceeded to calculate pathway-specific ratios using Shorthouse et al. dataset as per our manuscript. We verified that there were 5 optimum clusters using gap statistics, elbow method and silhouette method, consistent with the results generated using our metabolite ratios.
 - a. We observed distinct metabolite patterns among ratios expressed relative to proline and glutamine, but not to pyruvate (Figure for Reviewer 3), which we identified in our metabolite concentration dataset.

- b. Members of the clusters differed from ours, although there were some overlaps (Table for Reviewer 1). Our Cluster 3 spans Shorthouse's Cluster 1 and 4, and our Cluster 2 and 5 spans Shorthouse's Cluster 3 and 5.
- c. This lack of consistency between the analysis of the Shorthouse dataset and ours is likely explained by the negative relationship identified in Step 1.

Figure for Reviewer 3. Heatmap of scaled metabolite ratios by pathways covered in the untargeted metabolomics approach by Shorthouse et al. Ratios are calculated for all metabolites of a specific pathway against a precursor metabolite.

Table for Reviewer 1: Overlapping cell lines and cluster membership identified using pathway-ratio analyses.

Cluster #	Ours	Shorthouse	Cluster #
1	MCF7	MCF7	4
3	NICH226	NICH226	4
3	A549	A549	4
3	DU145	MIAPACA2	4
3	BT20	BXPC3	1
3	OVCAR3	OVCAR3	1
3	PC3	PC3	1
4	MDAMB468	MDAMB468	2
2	MIAPACA2	BT20	2
2	HCT116	HCT116	5
2	HT29	HT29	3
2	MDAMB231	MDAMB231	3
5	ASPC1	ASPC1	3
5	PANC1	PANC1	3
5	BXPC3	DU145	3

Bold cells indicate pairs of cell lines clustered together in both ours and Shorthouse. Rows are ordered to align common cell lines between the two datasets without splitting members of a cluster.

5. Since we did not observe identical clustering of the 15 common cell lines when using pathway-specific ratios of 39 metabolites, we attempted to directly test whether the differences in TCA cycle metabolites relative to pyruvate that we identified in our Cluster 4 cell lines and Cluster 3 (i.e. Figure 3C) was also present in the calculations when using the Shorthouse dataset.
 - a. There was no difference between these two groups using this approach (Figure for Reviewer 4).
 - b. This is almost certainly due to the small number of the 15 overlapping cell lines being members of Clusters 3 (6 cell lines) and 4 (1 cell line) (Table for Reviewer 1).
 - c. Therefore, we could not take this alternative course.

Figure for Reviewer 4: Abundance ratios of TCA cycle metabolites to pyruvate calculated from Shorthouse using the clusters identified in our study (C4 n=1, C3 n=6, data reported as individual values and mean).

The bigger picture is that these analyses support our overall conclusions that more meaningful differences are identified using pathway-centric ratios compared to metabolite abundance alone. However, including these analyses in our revised manuscript is challenging since **1)** there are many negative correlations between our metabolite concentration values and the abundance values reported in Shorthouse, and **2)** the resulting composition of the clusters generated using the ratios calculated from the abundance data of Shorthouse are very different to what we identified using our metabolite concentration data.

As such, we see two options for these analyses: **1)** do not include in the manuscript as the conflict in the results will be confusing to the reader and will require significant explanation for the possible reasons why, or **2)** include and highlight the bigger picture benefits of our approach, and attempt to avoid deep discussion on why there are differences in the cell line composition of the clusters. We prefer not to include these analyses (hence why they are not included in our revised manuscript), but we are very interested in the views of the Reviewer and Editor.

2. On statistical tests, the purpose of multiple test correction is to avoid false positives that can arise from performing large numbers of tests, and is not solely required when doing comparisons. This is important here as some of the results appear borderline significant and adjusting the threshold will reduce the risk of this type of error.

Response: We re-analysed the relevant data using FDR set at 5% and have included text in the appropriate locations and updated the figures in the revised manuscript.

3. Finally, the analysis of clustering remains inadequate, and the presented data does not clearly enable the authors to make the argument that five clusters is a more meaningful split than four or seven. Whilst the authors do not need to use sigclust, they need to better statistically justify the clusters selected as they are fundamental to the paper.

Response: We appreciate the reviewer's point that the justification of the clusters is important. It was an oversight that sufficient information was not included in the previous version of the manuscript. We have included additional text explaining the analyses that identified 4 clusters for the metabolite dataset (lines 85-90) and 5 clusters for the metabolite ratios (lines 119-124), plus in the Methods section (lines 411-414). We have also added the supporting data in Figure EV1B in our revised manuscript, which is also presented here as Figure for Reviewer 5.

Briefly, it is widely adopted that the optimal number of clusters can be determined based on three approaches: elbow method (Thorndike, 1953), average silhouette method (Rousseeuw, 1987) and gap statistics method (Tibshirani *et al*, 2002). We showed the outcomes of elbow method and silhouette method in our previous response. We now show the results from the gap statistic alongside those from the elbow and silhouette methods (Figure for Reviewer 5). Gap statistic is a statistical method by Tibshirani *et al.* (2002) that compares the total within intra-cluster variation with the variation under a null (random uniform) distribution. The larger the gap statistic, the better, as it indicates the clustering pattern is more different from a random distribution. The gap statistic shows that while a single cluster yields the highest statistic, the range between 3 to 7 clusters also yields similar statistics. In conclusion, all three methods show 3 to 7 clusters are reasonable. Combined with our domain knowledge of metabolite functions, and evidence of function validation differences between clusters, we believe that 5 clusters best represent the biological pattern in our data.

Figure for Reviewer 5: Line plots showing **A.** the gap statistic, **B.** the total within sum of squares using the Elbow method, and **C.** the average silhouette width for different number of clusters (x-axis).

Reviewer #2:

The authors have considered all points raised and have carefully revised the manuscript satisfactorily.

Just a final comment that is not intended to hinder publication of this work - the authors have argued well why they think measurement error likely has a minor impact on metabolite ratios. However, I would have liked finding such a comment also in the manuscript, to make the reader aware that it could play a role if one were to adapt the approach to other data.

Response: We thank the reviewer for their support. The suggestion to include our view on measurement error and metabolite ratios is very fair and a logical addition to the revised manuscript. We have added a couple of sentences to the limitations paragraph in the Discussion (lines 312-324).

Reviewer #3:

The manuscript is clear and well-written, and the additional details provide important insight for the reader. The authors clearly demonstrate the utility and conceptual innovation of how metabolomic data transformations can gain new insights into cancer cell biology.

I want to thank the authors for their efforts in addressing these critiques and for their thoughtful and thorough responses. I have no further comments.

Response: Thank you for your support.

References for Reviewers

Cherkaoui S, Durot S, Bradley J, Critchlow S, Dubuis S, Masiero MM, Wegmann R, Snijder B, Othman A, Bendtsen C *et al* (2022) A functional analysis of 180 cancer cell lines reveals conserved intrinsic metabolic programs. *Mol Syst Biol* 18: e11033

Golikov MV, Valuev-Elliston VT, Smirnova OA, Ivanov AV (2022) Physiological Media in Studies of Cell Metabolism. *Mol Biol* 56: 629-637

Rousseeuw PJ (1987) Silhouettes: A graphical aid to the interpretation and validation of cluster analysis. *Journal of Computational and Applied Mathematics* 20: 53-65

Shorthouse D, Bradley J, Critchlow SE, Bendtsen C, Hall BA (2022) Heterogeneity of the cancer cell line metabolic landscape. *Mol Syst Biol* 18: e11006

Thorndike RL (1953) Who belongs in the family? *Psychometrika* 18: 267-276

Tibshirani R, Walther G, Hastie T (2002) Estimating the Number of Clusters in a Data Set Via the Gap Statistic. *Journal of the Royal Statistical Society Series B: Statistical Methodology* 63: 411-423

11th Mar 2025

Manuscript Number: MSB-2024-12275RR

Title: Pathway metabolite ratios reveal distinctive glutamine metabolism in a subset of proliferating cells

Dear Dr Hoy,

Thank you for the submission of your revised manuscript to Molecular Systems Biology. I am pleased to inform you that we will be able to accept your manuscript pending the following final amendments and appropriate response to reviewers:

- 1) In the main manuscript file, please reduce keywords to max. 5.
- 2) Please format the Data availability section according to the example below:
"The datasets and computer code produced in this study are available in the following databases:
- Chip-Seq data: Gene Expression Omnibus GSE46748 (<https://www.ncbi.nlm.nih.gov/geo/query/acc.cgi?acc=GSE46748>)
- Modeling computer scripts: GitHub (<https://github.com/SysBioChalmers/GECKO/releases/tag/v1.0>)
- [data type]: [full name of the resource] [accession number/identifier] ([doi or URL or identifiers.org/DATABASE:ACCESSION])"
- 3) Data availability: The metabolomics datasets in Github are not publicly available. Please be aware that all deposited datasets should be freely accessible prior to publication.
- 4) Author contributions: Please remove it from the manuscript and specify author contributions in our submission system. CRediT has replaced the traditional author contributions section because it offers a systematic machine-readable author contributions format that allows for more effective research assessment. You are encouraged to use the free text boxes beneath each contributing author's name to add specific details on the author's contribution. More information is available in our guide to authors:
<https://www.embopress.org/page/journal/17574684/authorguide#authorshipguidelines>
- 5) Data citations: Please include a resolvable link to the landing page from which the data can be accessed at the end of each data citation reference. URLs for data citations (Data ref: Corsello et al., 2020), (Data ref: Yang et al., 2012), (Data ref: Tsherniak et al., 2017), (Data ref: Behan et al., 2019) are missing.
- 6) In the Methods, please take care of the following:
 - The Materials and Methods section should be renamed to "Methods".
 - Please ensure that a statement on whether or not blinding was done is included in the Methods even if no blinding was done. Please also be sure to update the Author Checklist with this information and where it can be found in the manuscript.
- 7) Please remove the Reagents and Tools Table from the Methods section of the manuscript and upload it as a separate file choosing the file type "Reagent Table".
- 8) Please place individual sections of the manuscript in the following order: Title page - Abstract & Keywords - Introduction - Results - Discussion - Methods - Data Availability - Acknowledgements - Disclosure and Competing Interests Statement - References - Figure Legends - Expanded View Figure Legends.
- 9) For the figures and figure legends, please take care of the following:
 - Please note that the exact p values are not provided in the legends of figures 3c, e-g; 4c; EV 2f-g.
 - Please note that the box plots need to be defined in terms of minima, maxima, centre, bounds of box and whiskers, and percentile in the legends of figures 3c, e-f; EV 2a.
 - Please note that information related to n is missing in the legends of figures EV 2a.
 - Although 'n' is provided, please describe the nature of entity for 'n' in the legends of figures 3c, e-f.
- 10) Tables: Appendix Table 1 should be renamed to Table EV1 with the corresponding callout in the main manuscript. Appendix Table 2 should be renamed to Dataset EV1 with the corresponding callout and the legend included as a separate tab in the Excel file.
- 11) Appendix: There is a callout in the main manuscript for an Appendix Figure S1, but no such figure or Appendix has been uploaded as part of your submission.
- 12) Funding: Please ensure that all funding sources are entered into the manuscript submission system (i.e. please add the Robinson Fellowship)
- 13) As part of the EMBO Publications transparent editorial process initiative (see our policy here: https://www.embopress.org/transparent-process#Review_Process), Molecular Systems Biology will publish online a Peer Review File (PRF) to accompany accepted manuscripts. This file will be published in conjunction with your paper and will include the anonymous referee reports, your point-by-point response and all pertinent correspondence relating to the manuscript. Let us know whether you agree with the publication of the PRF and as here, if you want to remove or not any figures from it prior to publication. Please note that the Authors checklist will be published at the end of the PRF.
- 14) After your paper is published, we will promote it on social media. If you have any handles or hashtags for Bluesky you would like included, please let us know.
- 15) Please provide a point-by-point letter INCLUDING my comments as well as the reviewer's reports and your detailed responses (as Word file).

I look forward to reading a new revised version of your manuscript as soon as possible.

Yours sincerely,

Poonam Bheda, PhD
Scientific Editor
Molecular Systems Biology

Reviewer #1:

Thanks to the authors for a thorough update of the manuscript. The authors response and revision has addressed the major issues discussed in my previous review, and I'm happy to recommend publication.

As the authors asked whether the substantial new analysis of Shorthouse/Cherkaoui should be included in the main text of the manuscript, I would argue that it should be included, though I accept that it complicates the narrative. Fundamentally the reanalysis talks to both the reproducibility of the results, and their biological importance. That they have shown similar (though distinct) clustering is reassuring and speaks to the robustness of their approach. With respect to the biological importance, the choice to assess cell lines in the same media was made to draw out the cell line specific differences. As the authors note, the environment is likely to impact the results and as such the reanalysis draws out cell-type specific features. I think therefore that the additional analysis improves the paper impact and robustness and its a benefit to both the authors and the journal to include it.

We again thank the Reviewer and Editor for their time assessing our manuscript and for their constructive feedback.

In our latest revised manuscript, we have edited it as requested by the Editor and included most of our analyses of the data from Shorthouse et al. (2022), which we presented in our previous Response to Reviewers document.

A point-to-point response to the Editor and Reviewer's comments and requested changes are outlined below.

Editor:

1. In the main manuscript file, please reduce keywords to max. 5.

Response: This has been addressed.

2. Please format the Data availability section.

Response: This section has been formatted.

3. Data availability: The metabolomics datasets in Github are not publicly available. Please be aware that all deposited datasets should be freely accessible prior to publication.

Response: Apologies for the oversight; this has been rectified.

4. Author contributions. Please remove it from the manuscript and specify author contributions in our submission system.

Response: This section has been removed from the manuscript file, and the contributions finalised in the submission system

5. Data citations: Please include a resolvable link to the landing page from which the data can be accessed at the end of each data citation reference. URLs for data citations (Data ref: Corsello et al., 2020), (Data ref: Yang et al., 2012), (Data ref: Tsherniak et al., 2017), (Data ref: Behan et al., 2019) are missing.

Response: Links have now been added to both the Results and Methods text, where these references have been called out. The formatting is inspired by very recent articles and so hopefully this is correct. Apologies if not.

6. In the Methods, please take care of the following:

a) The Materials and Methods section should be renamed to "Methods".

Response: Addressed.

b) Please ensure that a statement on whether or not blinding was done is included in the Methods even if no blinding was done. Please also be sure to update the Author Checklist with this information and where it can be found in the manuscript.

Response: A statement has been added on line 550.

7. Please remove the Reagents and Tools Table from the Methods section of the manuscript and upload it as a separate file choosing the file type "Reagent Table".

Response: This has been completed, and the Reagent Table has been uploaded.

8. Please place individual sections of the manuscript in the following order: Title page - Abstract & Keywords - Introduction - Results - Discussion - Methods - Data Availability - Acknowledgements - Disclosure and Competing Interests Statement - References - Figure Legends - Expanded View Figure Legends.

Response: Addressed.

9. For the figures and figure legends, please take care of the following:

a) Please note that the exact p values are not provided in the legends of figures 3c, e-g; 4c; EV 2f-g.

Response: This has been addressed and completed in line with guidance from the Editor

b) Please note that the box plots need to be defined in terms of minima, maxima, centre, bounds of box and whiskers, and percentile in the legends of figures 3c, e-f; EV 2a.

Response: This has been addressed for all legends related to box and whiskers plots.

c) Please note that information related to n is missing in the legends of figures EV 2a.

Response: This information has been included.

d) Although 'n' is provided, please describe the nature of entity for 'n' in the legends of figures 3c, e-f.

Response: We have clarified 'n' for all figure legends where it had been insufficiently defined.

10. Tables: Appendix Table 1 should be renamed to Table EV1 with the corresponding callout in the main manuscript. Appendix Table 2 should be renamed to Dataset EV1 with the corresponding callout and the legend included as a separate tab in the Excel file.

Response: This change has been made.

11. Appendix: There is a callout in the main manuscript for an Appendix Figure S1, but no such figure or Appendix has been uploaded as part of your submission.

Response: Apologies for the confusion and labelling error. This has been corrected.

12. Funding: Please ensure that all funding sources are entered into the manuscript submission system (i.e. please add the Robinson Fellowship).

Response: This information has been added.

13. As part of the EMBO Publications transparent editorial process initiative (see our policy here: https://www.embopress.org/transparent-process#Review_Process), Molecular Systems Biology will publish online a Peer Review File (PRF) to accompany accepted manuscripts. This file will be published in conjunction with your paper and will include the anonymous referee reports, your point-by-point response and all pertinent correspondence relating to the manuscript. Let us know whether you agree with the publication of the PRF and as here, if you want to remove or not any figures from it prior to publication. Please note that the Authors checklist will be published at the end of the PRF.

Response: We are supportive of this initiative.

14. After your paper is published, we will promote it on social media. If you have any handles or hashtags for Bluesky you would like included, please let us know.

Response: @hoylipidlab.bsky.social, @nancyts.bsky.social,
@sydneyprecisionds.bsky.social, @sydneyuni.bsky.social

Reviewer #1:

Thanks to the authors for a thorough update of the manuscript. The authors response and revision has addressed the major issues discussed in my previous review, and I'm happy to recommend publication.

As the authors asked whether the substantial new analysis of Shorthouse/Cherkaoui should be included in the main text of the manuscript, I would argue that it should be included, though I accept that it complicates the narrative. Fundamentally the reanalysis talks to both the reproducibility of the results, and their biological importance. That they have shown similar (though distinct) clustering is reassuring and speaks to the robustness of their approach. With respect to the biological importance, the choice to assess cell lines in the same media was made to draw out the cell line specific differences. As the authors note, the environment is likely to impact the results and as such the reanalysis draws out cell-type specific features. I think therefore that the additional analysis improves the paper impact and robustness and its a benefit to both the authors and the journal to include it.

Response: We thank the reviewer for their support for the publication of our manuscript. In our revised manuscript, we have included most of our analyses of Shorthouse et al that we documented in our last response to reviewer comments. We have added the metabolite clustering, pathway ratio clustering and resulting clusters of cell lines at the end of the Results section (lines 241-288) and have created a new Figure 5 and associated Figure EV4 plus Table EV3. Overall, we have taken the approach to highlight the more significant insights that can be achieved using pathway-centric metabolite ratio analyses and avoided deep comparisons between the results from our metabolomic data and that of Shorthouse and colleagues.

27th Mar 2025

Manuscript number: MSB-2024-12275RRR

Title: Pathway metabolite ratios reveal distinctive glutamine metabolism in a subset of proliferating cells

Dear Dr Hoy,

Thank you again for sending us your revised manuscript. We are now satisfied with the modifications made and I am pleased to inform you that your paper has been accepted for publication.

Yours sincerely,

Sincerely,

Poonam Bheda, PhD
Scientific Editor
Molecular Systems Biology
